# One Intervention per Component is Enough:
# Towards Identifiability in Linear Stochastic Dynamics from Steady State

**Saber Salehkaleybar** [1]

## Abstract

We study the problem of recovering the parameters of a multivariate Ornstein–Uhlenbeck (OU) process from steady-state observational and interventional data. In many applications, such as large-scale gene perturbation experiments, only stationary "snapshot" measurements are available, making standard stochastic differential equation estimation methods that rely on time-series trajectories inapplicable. We first establish an identifiability result: one intervention per strongly connected component (SCC) of the drift graph suffices to recover all OU process parameters generically up to a global scaling factor. This holds provided that the SCC condensation graph is connected with a single root and certain spectral nondegeneracy assumptions hold. We propose a recursive learning algorithm that orders SCCs topologically and, for each component, isolates its marginal dynamics and solves a linear system derived from the steady-state moment equations, leveraging parameters recovered for upstream components. Building on this theoretical foundation, we propose a regularized least-squares estimator that jointly minimizes residuals of the steady-state mean and covariance equations across observational and interventional data. Experimental results validate our theoretical findings in recovering parameters of the underlying OU process.

## 1. Introduction

Understanding the structure and parameters of dynamical systems from data is a fundamental problem in many scientific domains, including systems biology (Bansal et al.,

2006; Marbach et al., 2012), neuroscience (Paninski et al., 2010), and economics (Hamilton, 2020). In many settings, such as gene regulatory networks, the underlying dynamics can be modeled by a stochastic differential equation (SDE) whose steady-state distribution encodes both the interaction structure and kinetic parameters (Gardiner, 2009; Villaverde et al., 2016). Accurately recovering these parameters enables causal representation of interactions among variables, and predictions under unseen perturbations (Pearl, 2009). However, in experimental biology and related areas, data are often available only in the form of "snapshot" measurements (Cao et al., 2019; Schiebinger et al., 2019). The time-series trajectories (required by many SDE estimation methods (Zhang et al., 2024; Oh et al., 2024)) are expensive or infeasible to obtain at scale. In such cases, interventions such as gene knockdown (Datlinger et al., 2017) provide additional information by selectively intervening on parts of the system, potentially resolving non-identifiability issues that arise from observational data alone (Peters et al., 2017).

A substantial body of work addresses causal inference in linear stochastic systems. (Varando & Hansen, 2020; Dettling et al., 2024) focus on recovering drift structures from steady-state data using the Lyapunov equation. These approaches rely solely on observational measurements and do not leverage interventions, which can limit identifiability. More recently, (Lorch et al., 2024), focused on fitting the observational/interventional distributions without providing parameter recovery guarantees. Intervention-aware models, such as in (Rohbeck et al., 2024), achieve good empirical prediction, but the identifiability results are very restricted (see the related work section for more details).

In this paper, our aim is to incorporate interventions into the learning process, establishing identifiability results on recovering the parameters of SDEs and developing learning algorithms grounded in these theoretical results. Our contributions are as follows:

- We establish an identifiability result for recovering the parameters of a multivariate Ornstein–Uhlenbeck (OU) process from steady-state observational and interventional data. We show that a single intervention per SCC of the drift graph is sufficient to recover the parame-

---

[1]Leiden Institute of Advanced Computer Science, Leiden University, The Netherlands. Correspondence to: Saber Salehkaleybar <s.salehkaleybar@liacs.leidenuniv.nl>.

*Proceedings of the 43rd International Conference on Machine Learning*, Seoul, South Korea. PMLR 306, 2026. Copyright 2026 by the author(s).

ters up to a global scaling[1] if the SCC condensation graph is connected with a single root and certain spectral/rank nondegeneracy assumptions (Assumptions A.1 and A.2) hold. To prove this, we first consider the case where the drift matrix is a single SCC and show that by solving the linear system of equations for the first and second moments (based on observational data and data from one intervention), we can recover all the parameters up to a global scaling (Theorem 3.1).

- For the general case with multiple SCCs (Theorem 3.6), we provide a recursive algorithm that topologically orders the SCCs and, for each component, isolates its marginal dynamics. This reduces the problem to a sequence of single-SCC cases. For each SCC, the algorithm solves a linear system (derived from the steady-state moment equations for the observational and interventional settings) using parameters recovered for upstream components. We also provide an example that if the SCC condensation graph has more than one root, then it is impossible to learn all the parameters with a single global scaling.

- We show that changes in the steady-state means under single-node interventions reveal the SCC-level structure of the graph. In particular, Theorem 3.3 characterizes the set of nodes whose means change after an intervention as the downstream region of the intervened SCC. Consequently, one intervention per SCC is sufficient to recover the SCC decomposition and a topological order of the DAG over SCCs (Remark 3.4). This structural recovery step does not require the spectral/rank nondegeneracy assumptions used for parameter identification.

- Building on these theoretical findings, we formulate a regularized least-squares optimization problem that jointly minimizes the squared residuals of the steady-state mean and covariance equations across both observational and interventional data. Empirical results validate the identifiability results in recovering parameters and predicting unseen interventions.

## 2. Problem Formulation

**Ornstein–Uhlenbeck process**: The Ornstein–Uhlenbeck (OU) process is a continuous-time stochastic process that satisfies the stochastic differential equation (SDE):

$$d\mathbf{x} = (-\mathbf{\Lambda}\mathbf{x} + \mathbf{b})\,dt + \boldsymbol{\sigma}\,d\mathbf{W}_t, \tag{1}$$

where $\mathbf{x} \in \mathbb{R}^n$ is the state vector, $\mathbf{\Lambda} \in \mathbb{R}^{n \times n}$ is a drift matrix (assumed positive stable), $\mathbf{b} \in \mathbb{R}^n$ is a constant vector

representing external input, $\boldsymbol{\sigma} \in \mathbb{R}^{n \times n}$ is the diffusion matrix, and $d\mathbf{W}_t$ is an $n$-dimensional Wiener process.

The stationary distribution of the OU process has a multivariate normal distribution with the following mean and covariance:

- Mean:

$$\boldsymbol{\mu} = \mathbb{E}[\mathbf{x}_\infty] = \mathbf{\Lambda}^{-1}\mathbf{b}, \tag{2}$$

  where $\mathbf{x}_\infty$ denotes a random vector distributed according to the stationary distribution of the OU process.

- Covariance:

$$\mathbf{\Lambda}\mathbf{\Sigma} + \mathbf{\Sigma}\mathbf{\Lambda}^\top = \boldsymbol{\sigma}\boldsymbol{\sigma}^\top, \tag{3}$$

  where $\mathbf{\Sigma} \in \mathbb{R}^{n \times n}$ is the steady-state covariance matrix: $\mathbf{\Sigma} = \mathbb{E}[(\mathbf{x}_\infty - \boldsymbol{\mu})(\mathbf{x}_\infty - \boldsymbol{\mu})^\top]$.

We assume that the diffusion power matrix is diagonal and positive: $\mathbf{D} := \boldsymbol{\sigma}\boldsymbol{\sigma}^\top = \mathrm{diag}(d_1, \ldots, d_n) \succ 0$.

**Intervention:** We define an intervention on $i$-th coordinate of $\mathbf{x}$ as follows, where the $i$-th row of the drift matrix $\mathbf{\Lambda}$ is modified to remove influence from all other variables, while preserving its self-regulation. Formally, we define the modified drift matrix $\widetilde{\mathbf{\Lambda}}^{(i)}$ as:

$$\widetilde{\Lambda}_{kj}^{(i)} = \begin{cases} \Lambda_{kj}, & i \neq k, \\ 0, & i = k,\ j \neq i, \\ \Lambda_{ii}, & i = k,\ j = i. \end{cases}$$

That is, we zero out all off-diagonal entries in the $i$-th row, but retain $\Lambda_{ii}$, analogous to a knockout perturbation that isolates a variable from its regulators. This notion of intervention is often considered in the causal SDE literature, see, e.g., (Hansen & Sokol, 2014) and (Boeken & Mooij, 2024), where some post-intervention SDEs are formulated in this form.

The modified dynamic under intervention is: $d\mathbf{x} = (-\widetilde{\mathbf{\Lambda}}^{(i)}\mathbf{x} + \mathbf{b})\,dt + \boldsymbol{\sigma}\,d\mathbf{W}_t$. In the intervened system, the mean is denoted by $\boldsymbol{\mu}^{(i)}$ and satisfies:

$$\boldsymbol{\mu}^{(i)} = \mathbb{E}[\mathbf{x}_\infty^{(i)}] = \left(\widetilde{\mathbf{\Lambda}}^{(i)}\right)^{-1}\mathbf{b}, \tag{4}$$

assuming $\widetilde{\mathbf{\Lambda}}^{(i)}$ is invertible.

Similarly, the steady-state covariance $\mathbf{\Sigma}^{(i)}$ satisfies the Lyapunov equation:

$$\widetilde{\mathbf{\Lambda}}^{(i)}\mathbf{\Sigma}^{(i)} + \mathbf{\Sigma}^{(i)}\left(\widetilde{\mathbf{\Lambda}}^{(i)}\right)^\top = \boldsymbol{\sigma}\boldsymbol{\sigma}^\top. \tag{5}$$

We assume that, for every intervention considered, $\widetilde{\mathbf{\Lambda}}^{(i)}$ is positive stable, so the interventional stationary mean and covariance are well-defined.

---

[1]In OU processes, from the stationary distribution, parameters are identifiable at most up to a global scaling, an intrinsic ambiguity that cannot be resolved from stationary data alone.

Our goal is to recover the parameters of OU process, i.e., $\mathbf{\Lambda}, \mathbf{b}$, and $\boldsymbol{D}$ from the observational mean and covariance $(\boldsymbol{\mu}, \boldsymbol{\Sigma})$ and a collection of interventional means and covariances $\{(\boldsymbol{\mu}^{(i)}, \boldsymbol{\Sigma}^{(i)})\}_{i \in \mathcal{I}}$ where $\mathcal{I}$ is the set of coordinates intervened on. Please note that the parameters are identifiable only up to a global scaling because the stationary first-moment and second-moment equations are invariant under global scaling of parameters.

**Graph definitions:** In the following, we briefly review the graph-theoretic concepts used throughout the paper:
- **Strongly connected components.** For a directed graph $G$, a *strongly connected component (SCC)* is a maximal subset of nodes $C$ such that for every pair $u, v \in C$ there exists a directed path from $u$ to $v$ and a directed path from $v$ to $u$. The SCCs of $G$ form a partition of its nodes.
- **Condensation graph.** Given the SCCs $C_1, \ldots, C_K$ of $G$, the *condensation graph* is a directed graph whose nodes correspond to the SCCs and which contains a directed edge $C_i \to C_j$ whenever there exists an edge in $G$ from some node in $C_i$ to some node in $C_j$. By construction, the condensation graph is always a directed acyclic graph (DAG).
- **Topological order over SCCs.** A *topological order* of the SCCs is any ordering of the nodes of the condensation graph such that all directed edges point from earlier to later components. Equivalently, $C_i$ may appear before $C_j$ in the order if and only if there is no directed path from $C_j$ to $C_i$ in the condensation graph.

## 3. Identifiability Results

For a drift matrix $\mathbf{\Lambda}$, define the associated directed graph $G(\mathbf{\Lambda})$ by considering an edge $j \to i$ if and only if $\Lambda_{ij} \neq 0$. In the following, first we consider that $G(\mathbf{\Lambda})$ is strongly connected. Under some spectral/rank non-degeneracy assumptions, we show that the parameters of the OU process can be recovered generically up to some global scaling by just having one intervention on any coordinate. We then consider the case where $G(\mathbf{\Lambda})$ has multiple SCCs. In this setting, we first show that changes in interventional means identify the SCCs and a topological ordering over the SCC condensation DAG[2]. Given this SCC-level structure, we prove that the parameters are generically identifiable up to a single global scaling, provided that the SCC condensation DAG is connected, has a unique root SCC, and there is at least one intervention in each SCC.

### 3.1. Single SCC

**Theorem 3.1.** *Consider the OU process in equation 1 with true parameters $\mathbf{\Lambda}, \mathbf{b}$, and $\boldsymbol{D}$. Moreover, suppose that we have access to the observational steady-state mean and*

---

[2]This structural step does not require the spectral/rank nondegeneracy assumptions.

*covariance $(\boldsymbol{\mu}, \boldsymbol{\Sigma})$ and an interventional mean and covariance $(\boldsymbol{\mu}^{(i)}, \boldsymbol{\Sigma}^{(i)})$ ($i$ can be any coordinate). If the graph $G(\mathbf{\Lambda})$ is strongly connected and certain spectral/rank nondegeneracy assumptions hold (See Assumption A.1 and Assumption A.2 in Appendix A.6), generically[3], any other parameter triple $(\widehat{\mathbf{\Lambda}}, \widehat{\mathbf{b}}, \widehat{\mathbf{D}})$ that yields the same observational and interventional moments must satisfy:*

$$\widehat{\mathbf{\Lambda}} = c\mathbf{\Lambda}, \quad \widehat{\mathbf{b}} = c\mathbf{b}, \quad \widehat{\mathbf{D}} = c\mathbf{D},$$

*for some scalar $c > 0$.*

All the proofs of theorems (if not given in the main body) are available in the appendix. The key idea in the proof is based on forming a system of linear equations according to equation 2, equation 3, equation 4, and equation 5, showing that any possible solution for this set of equations should be a scale of the true parameters. In particular, let

$$\Theta = \begin{bmatrix} \mathrm{vec}(\mathbf{\Lambda}) \\ \mathbf{b} \\ \mathbf{d} \end{bmatrix} \in \mathbb{R}^p,$$

where $p := n^2 + 2n$, and $\mathbf{d} := (d_1, \ldots, d_n)^\top$, where $d_k := D_{kk} > 0$. Moreover, the operator $\mathrm{vec}$ stacks the columns of its matrix argument. Therefore, $\mathrm{vec}(\mathbf{\Lambda}) \in \mathbb{R}^{n^2}$. Now, we can write the linear equations in the following form: $\mathbf{A}\,\Theta = \mathbf{0}$ where $\mathbf{A} \in \mathbb{R}^{m \times p}$ (refer to Appendix A.1 for the definition of $\mathbf{A}$) and $m = n^2 + 3n$. In the proof, we show that $\mathbf{A}$ has generically a one-dimensional null space under the conditions in the statement of the theorem.

*Remark* 3.2. The nondegeneracy assumptions used in the proofs include spectral/rank conditions on moment-derived linear systems. We do not prove their genericity in this work; instead, we evaluate them numerically in Appendix C. The precise statements of the assumptions and their roles in the proofs are deferred to Appendix A and Appendix C.2, respectively. Establishing genericity of these conditions alone is left as a direction for future work. For SCCs containing a directed cycle through all vertices, Appendix C.2 provides additional justification for the relevant injectivity condition.

### 3.2. General Case

In the general case, where the graph $G(\mathbf{\Lambda})$ has multiple SCCs, assume that the intervention set $\mathcal{I}$ contains at least one intervention targeting a variable inside each SCC. Moreover, the DAG of SCCs is connected and there is exactly one root SCC. Under these conditions and certain spectral/rank non-degeneracy assumptions (see Assumptions A.1 and A.2, in the appendix), we show that the parameters of the OU process can be recovered up to a global scaling. The key idea is to first recover a topological ordering over the

---

[3]Please see the definition of genericity in Appendix A.2.

SCCs by inspecting which means change under each intervention; this structural recovery step does not require the spectral/rank nondegeneracy assumptions. The following theorem allows us to identify the set of variables in each SCC, as well as a topological ordering over the SCCs. With this structural information in hand, we then recursively identify the model parameters for each SCC by conditioning on previously resolved components, thereby reducing the multi-SCC case to a sequence of single-SCC problems.

**Theorem 3.3.** *Let $C$ be an SCC in $G(\mathbf{\Lambda})$ and let $i \in C$. Consider the intervention on $i$ and let $\boldsymbol{\mu}$ and $\boldsymbol{\mu}^{(i)}$ be the observational and interventional steady-state means, respectively.*

*(Non-null case). If the $i$-th row is non-null, i.e., $\exists j \neq i$ with $\Lambda_{ij} \neq 0$, then, generically, for the set of parameters $(\mathbf{\Lambda}, \mathbf{b})$ consistent with $G(\mathbf{\Lambda})$,*

$$\mu_k^{(i)} \neq \mu_k \quad \Longleftrightarrow \quad k \in \mathrm{Desc}(C),$$

*where $\mathrm{Desc}(C)$ denotes the set of nodes in SCCs reachable from $C$ in the DAG of SCCs (including $C$ itself).*

*(Null case). If the $i$-th row has no off-diagonals, i.e., $\Lambda_{ij} = 0$ for all $j \neq i$, then $\boldsymbol{\mu}^{(i)} = \boldsymbol{\mu}$.*

*Remark* 3.4. For an intervention on $i \in C$, define $R_i := \{ j : \mu_j^{(i)} \neq \mu_j \}$. Suppose there is at least one intervention in each SCC. Based on the above theorem, intersecting and differencing the sets $R_i$ across interventions identifies the SCCs and yields a topological order for the DAG over SCCs (see Appendix A.3 for more details).

*Remark* 3.5. The same SCC-recovery argument extends to non-null soft row interventions, where the intervened drift differs from $\mathbf{\Lambda}$ only in row $i$, but the changed row is not necessarily obtained by hard zeroing. The set of coordinates whose means change is again generically the downstream region of the intervened SCC. The proof is given in Appendix A.4.

**Theorem 3.6.** *Consider the OU process in equation 1 with true parameters $\mathbf{\Lambda}, \mathbf{b}, \mathbf{D}$. Suppose we have access to the observational steady-state mean and covariance $(\boldsymbol{\mu}, \mathbf{\Sigma})$, as well as to interventional means and covariances $(\boldsymbol{\mu}^{(i)}, \mathbf{\Sigma}^{(i)})$ for at least one intervention in each SCC of $G(\mathbf{\Lambda})$. If the DAG over SCCs of $G(\mathbf{\Lambda})$ is connected with a single root SCC and certain spectral/rank non-degeneracy assumptions (see Assumption A.1 and A.2 in the appendix) hold, then, generically, any other parameter triple $(\widehat{\mathbf{\Lambda}}, \widehat{\mathbf{b}}, \widehat{\mathbf{D}})$ that yields the same observational and interventional moments must satisfy*

$$\widehat{\mathbf{\Lambda}} = c\mathbf{\Lambda}, \quad \widehat{\mathbf{b}} = c\mathbf{b}, \quad \widehat{\mathbf{D}} = c\mathbf{D},$$

*for some scalar $c > 0$.*

*Proof.* Based on Theorem 3.3, we can infer the SCCs and also a topological ordering over them if there is at least

one intervention in each SCC. Let us denote these SCCs based on the topological ordering as $C_1, C_2, \cdots, C_K$ where $K$ is the number of components. Suppose that we already learned the parameters of the OU process in the components $C_1, C_2, \cdots, C_r$ up to some global scaling. Now, we aim for learning the parameters in $C_{r+1}$.

We partition the state vector according to the SCC decomposition of $G(\mathbf{\Lambda})$:

$$\mathbf{x} = \underbrace{\mathbf{x}_P}_{C_1 \cup \cdots \cup C_r} \oplus \underbrace{\mathbf{x}_T}_{C_{r+1}} \oplus \underbrace{\mathbf{x}_F}_{C_{r+2} \cup \cdots \cup C_K},$$

where the operator $\oplus$ denotes concatenation of subvectors corresponding to disjoint index sets.

The steady-state mean and covariance matrices are partitioned accordingly:

$$\boldsymbol{\mu} = \begin{bmatrix} \boldsymbol{\mu}_P \\ \boldsymbol{\mu}_T \\ \boldsymbol{\mu}_F \end{bmatrix}, \qquad \mathbf{\Sigma} = \begin{bmatrix} \mathbf{\Sigma}_{PP} & \mathbf{\Sigma}_{PT} & \mathbf{\Sigma}_{PF} \\ \mathbf{\Sigma}_{TP} & \mathbf{\Sigma}_{TT} & \mathbf{\Sigma}_{TF} \\ \mathbf{\Sigma}_{FP} & \mathbf{\Sigma}_{FT} & \mathbf{\Sigma}_{FF} \end{bmatrix}.$$

Everything inside the $P$-block is assumed known, i.e., the blocks $\mathbf{\Lambda}_{PP}, \mathbf{b}_P$, and $\mathbf{D}_P$, up to the same scaling $c$.

Because $\mathbf{x}_P$ and $\mathbf{x}_T$ are jointly Gaussian, we can write:

$$\mathbf{x}_T = \mathbf{B}\mathbf{x}_P + \mathbf{r}, \quad \text{where } \mathbf{B} := \mathbf{\Sigma}_{TP}\mathbf{\Sigma}_{PP}^{-1}, \quad \mathbb{E}[\mathbf{r}\,\mathbf{x}_P^\top] = 0.$$

The regression matrix $\mathbf{B}$ is computable directly from the observed moments, with no dependence on model parameters. Moreover, the residual term $\mathbf{r} := \mathbf{x}_T - \mathbf{B}\mathbf{x}_P$ is a Gaussian variable with

$$\boldsymbol{\mu}_{T|P} = \boldsymbol{\mu}_T - \mathbf{B}\,\boldsymbol{\mu}_P, \qquad \mathbf{\Sigma}_{T|P} = \mathbf{\Sigma}_{TT} - \mathbf{\Sigma}_{TP}\mathbf{\Sigma}_{PP}^{-1}\mathbf{\Sigma}_{PT}. \tag{6}$$

Substituting the definitions of $\mathbf{B}$ and $\mathbf{r}$ into the OU dynamics yields:

$$\dot{\mathbf{r}} = -\mathbf{\Lambda}_{TT}\mathbf{r} + \left(-\mathbf{\Lambda}_{TP} - \mathbf{\Lambda}_{TT}\mathbf{B} + \mathbf{B}\mathbf{\Lambda}_{PP}\right)\mathbf{x}_P \\ + (\mathbf{b}_T - \mathbf{B}\mathbf{b}_P) + (\boldsymbol{\sigma}_T\dot{\mathbf{W}}_T - \mathbf{B}\boldsymbol{\sigma}_P\dot{\mathbf{W}}_P). \tag{7}$$

To identify the residual moment equations, we use the cross-covariance block of the Lyapunov equation:

$$\mathbf{\Lambda}_{TP}\mathbf{\Sigma}_{PP} + \mathbf{\Lambda}_{TT}\mathbf{\Sigma}_{TP} + \mathbf{\Sigma}_{TP}\mathbf{\Lambda}_{PP}^\top = \mathbf{0}.$$

Using $\mathbf{B} = \mathbf{\Sigma}_{TP}\mathbf{\Sigma}_{PP}^{-1}$, this gives

$$\mathbf{\Lambda}_{TP} + \mathbf{\Lambda}_{TT}\mathbf{B} = -\mathbf{B}\mathbf{\Sigma}_{PP}\mathbf{\Lambda}_{PP}^\top\mathbf{\Sigma}_{PP}^{-1}.$$

Combining this with the parent Lyapunov equation

$$\mathbf{\Lambda}_{PP}\mathbf{\Sigma}_{PP} + \mathbf{\Sigma}_{PP}\mathbf{\Lambda}_{PP}^\top = \mathbf{D}_P,$$

we obtain

$$\mathbf{\Lambda}_{TP} + \mathbf{\Lambda}_{TT}\mathbf{B} = \mathbf{B}\mathbf{\Lambda}_{PP} - \mathbf{B}\mathbf{D}_P\mathbf{\Sigma}_{PP}^{-1}. \tag{8}$$

For the mean, using

$$\mathbf{\Lambda}_{TP}\boldsymbol{\mu}_P + \mathbf{\Lambda}_{TT}\boldsymbol{\mu}_T = \mathbf{b}_T, \qquad \boldsymbol{\mu}_T = \boldsymbol{\mu}_{T|P} + \mathbf{B}\boldsymbol{\mu}_P,$$

together with equation 8, gives

$$\mathbf{\Lambda}_{TT}\boldsymbol{\mu}_{T|P} = \mathbf{b}_T - \mathbf{B}\left(\mathbf{b}_P - \mathbf{D}_P\mathbf{\Sigma}_{PP}^{-1}\boldsymbol{\mu}_P\right). \qquad (9)$$

Similarly, for the residual covariance,

$$\mathbf{\Lambda}_{TT}\mathbf{\Sigma}_{T|P} + \mathbf{\Sigma}_{T|P}\mathbf{\Lambda}_{TT}^{\top} = \mathbf{D}_T + \mathbf{B}\mathbf{D}_P\mathbf{B}^{\top}. \qquad (10)$$

Now, based on the first and second moments of the residual which are given above, we can identify the parameters of component $T$ and also $\mathbf{\Lambda}_{TP}$ up to the same scaling. In particular, we have:

$\mathbf{\Lambda}_{TT}$: Note that the corresponding diffusion power matrix of the residual moment equation is $\mathbf{D}_T + \mathbf{B}\mathbf{D}_P\mathbf{B}^{\top}$, and therefore it is not necessarily diagonal. Nevertheless, the proof of Theorem 3.1 can be adapted to recover $\mathbf{\Lambda}_{TT}$ up to the same scaling $c$ by solving the residual moment equations; see Appendix A.7.

$\mathbf{\Lambda}_{TP}$: Having $\mathbf{\Lambda}_{PP}$, $\mathbf{D}_P$, and $\mathbf{\Lambda}_{TT}$ up to the same scaling $c$, equation 8 gives

$$\mathbf{\Lambda}_{TP} = \mathbf{B}\mathbf{\Lambda}_{PP} - \mathbf{B}\mathbf{D}_P\mathbf{\Sigma}_{PP}^{-1} - \mathbf{\Lambda}_{TT}\mathbf{B}.$$

Therefore, $\mathbf{\Lambda}_{TP}$ is recovered with the same scaling.

$\mathbf{b}_T$ and $\mathbf{D}_T$: According to equation 2, we have

$$\mathbf{b}_T = \mathbf{\Lambda}_{TT}\boldsymbol{\mu}_T + \mathbf{\Lambda}_{TP}\boldsymbol{\mu}_P.$$

Since we recovered $\mathbf{\Lambda}_{TT}$ and $\mathbf{\Lambda}_{TP}$ with the same scaling factor, $\boldsymbol{b}_T$ is identifiable with the same scaling from the above equation.

Regarding $\boldsymbol{\sigma}_T$ (or diffusion power matrix $\mathbf{D}_T$), from the Lyapunov equation for the covariance matrix of residual (in other words, $\mathbf{\Sigma}_{T|P}$), we have:

$$\mathbf{\Lambda}_{TT}\mathbf{\Sigma}_{T|P} + \mathbf{\Sigma}_{T|P}\mathbf{\Lambda}_{TT}^{\top} = \mathbf{D}_T + \mathbf{B}\,\mathbf{D}_P\,\mathbf{B}^{\top}.$$

Therefore,

$$\mathbf{D}_T = \mathrm{diag}\left(\mathbf{\Lambda}_{TT}\mathbf{\Sigma}_{T|P} + \mathbf{\Sigma}_{T|P}\mathbf{\Lambda}_{TT}^{\top} - \mathbf{B}\mathbf{D}_P\mathbf{B}^{\top}\right),$$

and hence $\mathbf{D}_T$ is learned with the same scaling. This completes the recursive step and the proof. $\square$

*Remark* 3.7. The two structural assumptions in Theorem 3.6 are necessary. If the DAG over SCCs is disconnected, then each disconnected part of the DAG can only be identified up to its own scaling factor. Moreover, if there are multiple root SCCs, the parameters of the OU process cannot, in general, be recovered up to a single global scaling. An example illustrating this case is provided in Appendix A.8.

---

**Algorithm 1** Recursive OU Learning with Interventions

1: **Input:** Observational and interventional means and covariances $(\boldsymbol{\mu}, \mathbf{\Sigma}), \{(\boldsymbol{\mu}^{(i)}, \mathbf{\Sigma}^{(i)})\}_{i \in \mathcal{I}}$
2: **Phase 1: Learn SCCs**
3: Identify SCCs, $C_1, C_2, \ldots, C_K$, and a topological ordering over them from mean changes using Theorem 3.3
4: **Phase 2: Recover parameters per SCC**
5: **for** each component $T = C_j$ in topological order **do**
6:     Let $P := C_1 \cup \cdots \cup C_{j-1}$ be the union of previously processed SCCs
7:     Compute conditional mean $\boldsymbol{\mu}_{T|P}$ and covariance $\mathbf{\Sigma}_{T|P}$ according to equation 6
8:     Recover $\mathbf{\Lambda}_{TT}$ using the interventional moments with the same scaling in $P$
9:     Recover $\mathbf{\Lambda}_{TP}$, $\boldsymbol{b}_T$, $\mathbf{D}_T$ up to the same scaling using cross-covariances and stationary conditions
10: **end for**
11: **Output:** Drift matrix $\mathbf{\Lambda}$, input vector $\boldsymbol{b}$, and diffusion matrix $\boldsymbol{D}$ (up to a global scaling)

---

Building on the above, we design a recursive algorithm that proceeds in two stages (the pseudo-code is given in Algorithm 1):

- In the first stage, we use the changes in interventional means to identify the SCCs of the drift graph $G(\mathbf{\Lambda})$, along with a topological ordering over these components. This structural information is inferred using Theorem 3.3.

- In the second stage, we iterate over the SCCs in topological order. For each component $T$, we treat the union of previously processed SCCs as $P$, and condition on $\mathbf{x}_P$ to isolate the marginal dynamics of $\mathbf{x}_T$. Using the conditional moments $(\boldsymbol{\mu}_{T|P}, \mathbf{\Sigma}_{T|P})$, we first recover $\mathbf{\Lambda}_{TT}$ up to a scaling. Then, leveraging the structure of the OU dynamics, we recover $\mathbf{\Lambda}_{TP}$, $\boldsymbol{b}_T$, and $\mathbf{D}_T$ up to the same scaling. The procedure continues recursively until all components have been identified.

*Remark* 3.8. Rather than directly running Algorithm 1, one can alternatively construct a global linear system from the observational and interventional first- and second-moment equations, as in the proof of Theorem 3.1. If this system has a one-dimensional null space, then all parameters are identified up to a global scaling. Theorems 3.1 and 3.6 provide conditions under which this one-dimensional ambiguity is the only ambiguity, while Theorem 3.3 provides the SCC-level structural information used by the recursive procedure.

*Remark* 3.9. Although the parameters $(\mathbf{\Lambda}, \boldsymbol{b}, \boldsymbol{D})$ are identifiable only up to a global scaling, this ambiguity does not

affect the prediction of stationary moments. In particular, our identifiability result implies that any alternative parameter triple consistent with the observational and interventional steady state is of the form $\hat{\mathbf{\Lambda}} = c\mathbf{\Lambda}, \hat{b} = cb, \hat{D} = cD$, for some scalar $c > 0$. For the steady-state mean, we have $\boldsymbol{\mu} = \mathbf{\Lambda}^{-1}b$, and $\hat{\boldsymbol{\mu}} = \hat{\mathbf{\Lambda}}^{-1}\hat{b} = (c\mathbf{\Lambda})^{-1}(cb) = \mathbf{\Lambda}^{-1}b = \boldsymbol{\mu}$, so the scaling cancels out. The same holds for the steady-state covariance where $\mathbf{\Sigma}$ is defined as the unique solution of the Lyapunov equation $\mathbf{\Lambda\Sigma} + \mathbf{\Sigma\Lambda}^\top = D$, under the assumption that $\mathbf{\Lambda}$ is stable. Under the scaled parameters, the covariance $\hat{\mathbf{\Sigma}}$ satisfies $\hat{\mathbf{\Lambda}}\hat{\mathbf{\Sigma}} + \hat{\mathbf{\Sigma}}\hat{\mathbf{\Lambda}}^\top = \hat{D} \iff c\mathbf{\Lambda}\hat{\mathbf{\Sigma}} + \hat{\mathbf{\Sigma}}c\mathbf{\Lambda}^\top = cD \iff \mathbf{\Lambda}\hat{\mathbf{\Sigma}} + \hat{\mathbf{\Sigma}}\mathbf{\Lambda}^\top = D$. By uniqueness of the solution to the Lyapunov equation for a stable drift matrix, we obtain $\hat{\mathbf{\Sigma}} = \mathbf{\Sigma}$. The same arguments can be applied to an intervened drift $\mathbf{\Lambda}^{(i)}$, so the predicted moments for unseen interventions are also invariant to the global scaling.

## 4. Learning Algorithm

In finite-sample settings, plugging in empirical moments generally yields full-column-rank systems, which admit only the trivial solution of zero vector. Nevertheless, the identifiability result in Theorem 3.1 motivates replacing true moments with empirical ones and relaxing the equations into a least-squares objective. Specifically, in the observational case, the mean vector $\boldsymbol{\mu}$ and covariance matrix $\mathbf{\Sigma}$ satisfy the linear system in equation 2 and equation 3, respectively. Moreover, for each intervention $i \in \mathcal{I}$, the mean vector $\boldsymbol{\mu}^{(i)}$ and $\mathbf{\Sigma}^{(i)}$ satisfy equations in equation 4 and equation 5, respectively.

For the set of free parameters $\Theta = (\mathrm{vec}(\mathbf{\Lambda}), \mathbf{b}, \mathbf{d})$, where $\mathbf{d}$ is the diagonal of matrix $\mathbf{D}$, we define the following least-square objective:

$$
\begin{aligned}
\mathcal{L}(\Theta) = &\ \alpha_O \left( \left\| \mathbf{\Lambda}\,\hat{\boldsymbol{\mu}} - \mathbf{b} \right\|_2^2 + \left\| \mathbf{\Lambda}\,\hat{\mathbf{\Sigma}} + \hat{\mathbf{\Sigma}}\,\mathbf{\Lambda}^\top - \mathbf{D} \right\|_F^2 \right) \\
&+ \alpha_I \sum_{i \in \mathcal{I}} \left( \left\| \widetilde{\mathbf{\Lambda}}^{(i)}\,\hat{\boldsymbol{\mu}}^{(i)} - \mathbf{b} \right\|_2^2 \right) \\
&+ \alpha_I \sum_{i \in \mathcal{I}} \left( \left\| \widetilde{\mathbf{\Lambda}}^{(i)}\,\hat{\mathbf{\Sigma}}^{(i)} + \hat{\mathbf{\Sigma}}^{(i)}(\widetilde{\mathbf{\Lambda}}^{(i)})^\top - \mathbf{D} \right\|_F^2 \right),
\end{aligned}
\tag{11}
$$

where $\alpha_O, \alpha_I > 0$ are weighting coefficients; we often give a larger weight to observational terms since observational samples are often more abundant and their estimates are more accurate. Moreover, $\hat{\boldsymbol{\mu}}, \hat{\mathbf{\Sigma}}, \hat{\boldsymbol{\mu}}^{(i)}, \hat{\mathbf{\Sigma}}^{(i)}, i \in \mathcal{I}$ are the unbiased estimates of first and second moments in the observational and interventional settings and $\| \cdot \|_F$ is the Frobenius norm. To promote sparsity in the drift graph, we add an $\ell_1$ penalty on the off-diagonal elements of $\mathbf{\Lambda}$: $\mathcal{R}(\mathbf{\Lambda}) = \gamma \sum_{i \neq j} |\Lambda_{ij}|$, where $\gamma > 0$ controls the sparsity

level. Therefore, the optimization problem becomes

$$
\min_{\Theta}\ \mathcal{L}(\Theta) + \mathcal{R}(\mathbf{\Lambda}),
\tag{12}
$$

subject to stability constraints (such as $\Lambda_{kk} > 0$ and $d_k > 0$ for all $k$). Because the moment equations are homogeneous in $(\mathbf{\Lambda}, \mathbf{b}, \mathbf{D})$, one may impose a scale-fixing normalization, such as $\mathrm{tr}(\mathbf{D}) = 1$, to remove the global scaling ambiguity.

## 5. Related Work

Herein, we mainly review methods that perform inference from stationary distributions of dynamical systems, rather than from full trajectories[4].

**Identifiability and learning from steady state in SDE models.** A line of work on graphical continuous Lyapunov models treats the steady-state covariance as the solution of a continuous Lyapunov equation. In this setting, (Varando & Hansen, 2020) proposed an $l_1$-regularized estimator to recover sparse drift structure from observational snapshots. (Dettling et al., 2023) subsequently analyzed identifiability in this framework, proving that when only observational covariances are available and the diffusion matrix is known, the drift is globally identifiable from the covariance if and only if the drift graph is simple, meaning it contains no directed two-cycles. While these results provide valuable insights, they are limited to observational data, cannot recover models with two-cycles in the drift, and do not handle unknown diffusion. In contrast, our work incorporates interventional data, allows unknown diagonal diffusion, and shows that a single intervention in each SCC (under some conditions on the DAG of SCCs) suffices for recovery up to a global scaling. More recently, (Dettling et al., 2024) proposed a Lasso-based estimator for recovering the drift structure of linear SDEs from stationary observational covariances and analyzed its identifiability properties. However, their framework does not incorporate interventions. Very recently, (Zweig et al., 2025) studied identifiability of linear SDEs under mean-shift interventions, assuming a low-rank drift matrix. In contrast, we allow general drift matrices and base identifiability on the SCC structure of the drift graph rather than on a low rank assumption.

Other approaches aim to learn stationary SDE models without focusing on identifiability. (Lorch et al., 2024) proposed a kernel deviation-from-stationarity objective that measures how far a candidate SDE's stationary distribution deviates from the empirical distribution. Their framework can accommodate cycles and generalizes well to unseen interventions, but its goal is density fitting in reproducing kernel Hilbert spaces rather than moment-based parameter identification. Our work differs by deriving explicit graphical conditions

---

[4]There are several surveys on causal discovery from temporal data (e.g., (Gong et al., 2024; Hasan et al., 2023))

under which the OU parameters are recoverable from first and second moments.

Recent empirical work has also combined steady-state dynamics with interventions. (Rohbeck et al., 2024) introduced Bicycle, a model in which interventions alter a subset of parameters. Bicycle achieves good performance in both structure recovery and prediction under out-of-distribution interventions in single-cell datasets. However, it is designed primarily as a predictive model and provides identifiability guarantees only when interventions are performed on all coordinates except one. In contrast, we gave identifiability results under substantially weaker requirements, i.e., one intervention per SCC.

(Boege et al., 2025) have studied the conditional independence relations implied by sparsity in the drift of stationary multivariate diffusions. Their results link graph structure to conditional independencies in the stationary state distribution. Nonetheless, this line of work does not address the recovery of drift or diffusion parameters, nor how interventions could enable identifiability.

Finally, there exists related work for deterministic linear ODEs rather than stochastic OU models. For example, (Wang et al., 2024) analyzed the identifiability of linear ODE systems with hidden confounders from time series or discretely sampled trajectories and derived conditions under which latent confounding can be resolved.

**Gene perturbation prediction from steady state.** In systems biology, several methods aim to predict the effects of genetic perturbations directly from stationary data. For instance, (Sethuraman et al., 2023) proposed NODAGS-Flow, which learns nonlinear cyclic causal structures from interventional steady-state gene expression by fitting residual normalizing flows, producing predictions and plausible graphs. While providing good performance in practice, NODAGS-Flow is a likelihood-based method without formal identifiability guarantees on recovering the parameters of the underlying system.

Other works focus on high-performing predictive models. For instance, (Roohani et al., 2022) proposed GEAR, which combines graph neural networks with prior biological network information to predict transcriptional responses to single or multiple gene perturbations, showing improved generalization to unseen combinations. (Yu et al., 2025) proposed PerturbNet, which uses conditional invertible flows to model the distributional effects of unseen perturbations. PerturBench (Wu et al.) provides a unified benchmark suite for single-cell perturbation modeling, facilitating comparison between predictive models.

## 6. Experiments

**Scope of empirical evaluation.** The goal of this section is to validate the identifiability results (e.g., parameter recovery). Regarding comparisons, we focus on methods that assume linear SDE models and operate on stationary data. In particular, (Rohbeck et al., 2024) considered a Lyapunov-based loss that corresponds to our objective without the mean terms; we include this variant explicitly as an ablation (*Covariance only*). (Varando & Hansen, 2020) proposed an $\ell_1$-regularized estimator based solely on observational covariances (without interventions), and is therefore not designed for the interventional setting. (Lorch et al., 2024) also proposed a loss for linear SDEs, but their interventions are implemented as mean shifts rather than the hard interventions assumed in our work. We adapted their loss to our hard-intervention setting and evaluated it on synthetic data. However, we did not observe improvements in performance as the number of interventions increased, and hence did not report its results.

**Synthetic Data.** We generated synthetic datasets by simulating steady-state observations from a stable linear stochastic system. The drift matrix $\mathbf{\Lambda} \in \mathbb{R}^{n \times n}$ was initialized as a zero matrix, then each off-diagonal entry was set to a Gaussian random value with probability $\rho$ and left at zero otherwise, where $\rho$ is the desired density level. For each row, the diagonal entry was set to be larger than the sum of the absolute values of off-diagonal entries in that row by at least a positive margin, ensuring stability. The diffusion power $\mathbf{D}$ was diagonal with strictly positive entries sampled uniformly from $[d_{\min} = 0.2, d_{\max} = 0.4]$, and the bias vector $\mathbf{b}$ was drawn uniformly from $[b_{\min} = 0.2, b_{\max} = 1.5]$. The observational steady-state mean and covariance were then computed by solving the corresponding linear and Lyapunov equations for the given $\mathbf{\Lambda}$, $\mathbf{b}$, and $\mathbf{D}$. Interventions were simulated by zeroing all off-diagonal entries in a selected row of $\mathbf{\Lambda}$ while keeping the diagonal entry unchanged and leaving $\mathbf{b}$ unchanged. For the observational setting and each intervention, samples were drawn from the corresponding multivariate normal distribution. All the implementations are available in the following link: https://github.com/sabersalehk/OU_ID. More details of experiments are given in Appendix B.

In Figure 1 (a-c), we report the estimation error of $\mathbf{\Lambda}$, $\mathbf{D}$, and $\mathbf{b}$ as a function of the number of interventions with $n = 10$. To evaluate recovery up to scaling, we compute the scaling factor $c = \langle \hat{\mathbf{A}}, \mathbf{A} \rangle / \langle \mathbf{A}, \mathbf{A} \rangle$ for each parameter matrix/vector $\mathbf{A} \in \{\mathbf{\Lambda}, \mathbf{D}, \mathbf{b}\}$, where $\langle \mathbf{X}, \mathbf{Y} \rangle = \mathrm{tr}(\mathbf{X}^\top \mathbf{Y})$ for matrices and $\langle \mathbf{x}, \mathbf{y} \rangle = \mathbf{x}^\top \mathbf{y}$ for vectors. The relative error is then measured as $\|\hat{\mathbf{A}} - c\mathbf{A}\|/\|\mathbf{A}\|$. We apply this procedure separately to $\mathbf{\Lambda}$, $\mathbf{D}$, and $\mathbf{b}$. The results with 90% confidence intervals are given in blue curves with the legend "Mean and Covariance (90% CI)". As shown, for $\mathbf{\Lambda}$ and $\mathbf{b}$, the relative error decreases as the number of interventions

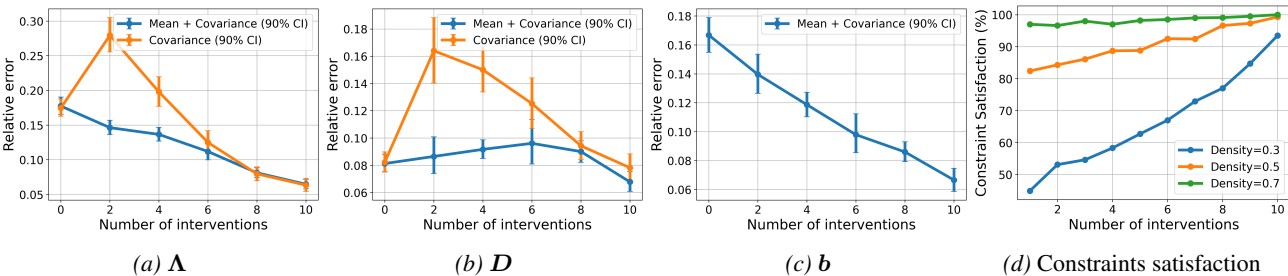

*(a) $\Lambda$*      *(b) $D$*      *(c) $b$*      *(d)* Constraints satisfaction

*Figure 1.* (a–c) Relative errors of estimated parameters of the OU process versus the number of interventions, and (d) percentage of instances satisfying graphical conditions in Theorem 3.6.

*Table 1.* Evaluation of DES and PDS on the three Perturb-seq datasets. The interventional mean is shaded to indicate oracle access to interventional data.

|  | Co-Culture | | Control | | IFN-$\gamma$ | |
|---|---|---|---|---|---|---|
| Method | DES | PDS | DES | PDS | DES | PDS |
| Observational mean | 0.33 | 0.57 | 0.33 | 0.57 | 0.35 | 0.67 |
| Ours ("Covariance") | 0.51 | 0.43 | **0.46** | 0.43 | 0.42 | 0.43 |
| Ours ("Mean + Covariance") | 0.43 | **0.67** | 0.33 | 0.63 | **0.47** | **0.70** |
| Interventional mean | **0.52** | 0.57 | 0.42 | **0.67** | 0.36 | **0.70** |

increases. For **D**, we observe a small non-monotone effect: the error is initially low with no interventions, increases slightly for a few interventions, and then decreases again. One possible explanation is that **D** is already estimated accurately from observational data, so incorporating a small number of interventional data (which may contain fewer samples than the observational data) can worsen its estimation. As the number of interventions increases, however, the estimation of $\Lambda$ improves, which in turn reduces the error in **D**.

We also report results when the loss includes only the residuals of the Lyapunov equations (i.e., using covariances but not means) with the legend "Covariance (90% CI)," similar to (Dettling et al., 2024; Rohbeck et al., 2024). This variant performs noticeably worse, highlighting that the mean terms are essential (please note that there is no curve for **b** in this case as there is no term for estimating it); mean estimates are typically much more accurate than covariance estimates, and incorporating them substantially improves recovery.

In Figure 1d, we depict the percentage of instances of $\Lambda$ satisfying the graphical conditions in Theorem 3.6 as a function of the number of interventions for different graph densities ($\rho$). For sparse graphs ($\rho = 0.3$), less than 50% of instances satisfy the conditions with a single intervention, but the percentage increases steadily as more interventions are added. In contrast, denser graphs ($\rho = 0.5, 0.7$) already satisfy the graphical conditions at a high rate with only a few interventions.

We also evaluated the SCC-recovery step from finite sam-

ples. Using the reconstruction procedure implied by Theorem 3.3 and Remark 3.4, we estimated the response sets from empirical mean shifts and recovered the SCC partition under both hard and soft interventions. We measured recovery quality using the Adjusted Rand Index (ARI), where 1 indicates exact recovery. As shown in Table 2, SCC recovery improves steadily as the number of samples increases. Hard interventions are more informative in this experiment, but soft interventions also show a clear improvement with sample size. The details of experiments are given in Appendix B.5.

*Table 2.* SCC recovery from estimated means. Recovery is measured by Adjusted Rand Index (ARI), where 1 indicates exact recovery.

| Sample size | Hard int. (ARI) | Soft int. (ARI) |
|---|---|---|
| 500 | 0.660 | 0.464 |
| 1000 | 0.746 | 0.615 |
| 5000 | 0.849 | 0.804 |
| 10000 | 0.881 | 0.837 |

Our identifiability theory assumes diagonal diffusion, and this assumption is mainly used for the parameter-identifiability result in Theorem 3.6. The learning objective, however, can also be applied with a non-diagonal diffusion matrix. To evaluate this empirically, we repeated the synthetic experiment in the same setup as Figure 1, but allowed the diffusion matrix to be non-diagonal. The results in Table 3 show the same qualitative trend as in the diagonal-diffusion setting: the relative errors for $\Lambda$, **D**, and

**b** decrease as the number of interventions increases. This suggests that, although the current identifiability proof uses diagonal diffusion, the proposed moment-based estimator remains empirically useful beyond that setting. The details of experiments are given in Appendix B.6.

*Table 3.* Parameter recovery with non-diagonal diffusion. Relative errors decrease as the number of interventions increases.

| # int. | Rel. err. ($\mathbf{\Lambda}$) | Rel. err. ($\mathbf{D}$) | Rel. err. ($\mathbf{b}$) |
|--------|--------|--------|--------|
| 0 | 0.2064 | 0.1811 | 0.1407 |
| 2 | 0.1782 | 0.1582 | 0.1423 |
| 4 | 0.1570 | 0.1449 | 0.1394 |
| 6 | 0.1198 | 0.1188 | 0.1185 |
| 8 | 0.0859 | 0.1011 | 0.0994 |
| 10 | 0.0646 | 0.0719 | 0.0661 |

We further empirically investigate how conservative the graphical conditions in Theorem 3.6 are in Appendix B.3, and study the impact of sample size on performance in Appendix B.4.

**Real Data.** We assess our method on real-world data, leveraging three published single-cell perturbation screen datasets (Frangieh et al., 2021). Since the true causal graph is unknown in this setting, our evaluation focuses on generalization to unseen perturbations. Specifically, we consider a Perturb-seq dataset containing targeted CRISPR knock-out perturbations of 249 target genes in tumor-infiltrating lymphocytes (TILs) of melanoma patients. The perturbations were performed under three conditions, which we treat as separate datasets: a baseline culture of TILs in a neutral medium ("Control"), a culture of TILs with interferon-$\gamma$ added ("IFN-$\gamma$"), and a co-culture of TILs with patient-derived melanoma cells ("Co-Culture").

Following the setup in (Sethuraman et al., 2023), we restrict our analysis to the same subset of 61 genes and adopt their reported training/test split: 90% of interventions are used for training and the remaining 10% are held out for evaluation, with analyses performed separately for each dataset. Our goal is to predict the interventional mean $\boldsymbol{\mu}$ for unseen perturbations, using the estimated parameters and the steady-state equation for the mean. Note that global scaling is not an issue here, as it cancels out for predicting $\boldsymbol{\mu}$.

For evaluation, we do not rely on Mean Absolute Error (MAE), as even the observational mean achieves a very close performance to the one using the interventional mean on the held-out set. Instead, following the recommendation in the Virtual Cell Challenge[5], we report the Differential Expression Score (DES) and the Perturbation Differential Score (PDS), where higher values indicate better performance.

DES measures agreement in identifying differentially expressed genes, while PDS measures a model's ability to distinguish between perturbations by ranking predictions according to their similarity to the true perturbational effect, regardless of their effect size. In Table 1, we compare our full method ("Mean + Covariance") against the "Covariance" only variant (similar to the approaches in (Dettling et al., 2024; Rohbeck et al., 2024)), and against the one using the interventional mean of the held-out set (which is not available to our method). As shown in Table 1, our method achieves comparable and in some cases even higher scores than the oracle baseline using the interventional mean, despite not having access to held-out interventional data.

## 7. Conclusions and Future Work

We studied recovery of multivariate OU parameters from steady-state observational and interventional data. Our main theoretical contribution shows that one intervention per SCC of the drift graph suffices for generic recovery of $(\mathbf{\Lambda}, \mathbf{b}, \mathbf{D})$ up to a single global scaling when the DAG over SCCs is connected with a unique root. The single-SCC case (Theorem 3.1) yields a rank-1 null space for the stacked moment equations, and the multi-SCC result (Theorem 3.6) follows via a constructive, recursive decomposition that leverages mean shifts (Theorem 3.3) to infer SCCs and a topological order. Building on these guarantees, we considered a regularized least-squares estimator and observed accurate parameter recovery in synthetic datasets.

While our theoretical results are developed for linear SDEs, we note that this setting is a canonical model class for studying causal inference in continuous-time systems where causal effects and parameter recovery can be analyzed rigorously. The identifiability results we obtain, can lead to some practical insights on when stationary data of interventions are sufficient to recover causal structure. This might be interesting for the wider ML community working on perturbation predictions.

Our guarantees rely on some spectral/rank nondegeneracy assumptions. Numerical results suggest these hold generically for strongly connected drift graphs. A key avenue for future work is a formal genericity proof. Another direction is to understand how diffusion assumptions (e.g., not diagonal diffusion matrices) change the boundary between identifiable and non-identifiable regimes.

## Impact Statement

This paper presents work whose goal is to advance the field of machine learning. There are many potential societal consequences of our work, none of which we feel must be specifically highlighted here.

---

[5]https://virtualcellchallenge.org/evaluation#scoring

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

# A. Proofs

## A.1. Forming the Linear System

Throughout, $\mathbf{I}_n$ denotes the $n \times n$ identity, $\otimes$ is the Kronecker product, and $\mathbf{e}_k$ is the $k$-th canonical basis vector in $\mathbb{R}^n$.

We define the following selector matrices:

- Introduce the matrix $\mathbf{P} \in \mathbb{R}^{n^2 \times n}$ such that $\mathrm{vec}\big(\mathrm{diag}(\mathbf{d})\big) = \mathbf{P}\,\mathbf{d}$, where the $k$-th column of $\mathbf{P}$ is given by $\mathbf{P}_{:,k} = \mathrm{vec}(\mathbf{E}_{kk})$, and $\mathbf{E}_{kk}$ denotes the $n \times n$ matrix with a one in entry $(k,k)$ and zeros elsewhere.

- For a fixed row index $i$, let $\mathbf{E}_i := \mathbf{I}_n \otimes \mathbf{e}_i^\top \in \mathbb{R}^{n \times n^2}$. Therefore, $\mathbf{E}_i \, \mathrm{vec}(\boldsymbol{\Lambda}) = \big[\Lambda_{i1}, \ldots, \Lambda_{in}\big]^\top$ extracts the entire $i$-th row of $\boldsymbol{\Lambda}$.

- $\mathbf{S}_n \in \mathbb{R}^{\frac{n(n+1)}{2} \times n^2}$ is the upper-triangular elimination matrix.

- Let $\mathbf{C} \in \mathbb{R}^{n^2 \times n^2}$ denote the commutation matrix, so that $\mathrm{vec}(\mathbf{X}^\top) = \mathbf{C} \, \mathrm{vec}(\mathbf{X})$ for any $X \in \mathbb{R}^{n \times n}$.

For the equation of observational mean, we define:

$$\mathbf{M}_0 := \left[ -\big(\boldsymbol{\mu}^\top \otimes \mathbf{I}_n\big) \,\middle|\, \mathbf{I}_n \,\middle|\, \mathbf{0} \right] \in \mathbb{R}^{n \times p},$$

where $p = n^2 + 2n$.

For the equation of the interventional mean (on coordinate $i$), let $\mathbf{J}_i := \mathbf{I}_n - \mathbf{e}_i \mathbf{e}_i^\top$. We define:

$$\mathbf{M}_1 := \left[ -\big((\boldsymbol{\mu}^{(i)})^\top \otimes \mathbf{I}_n\big) + \mathbf{e}_i(\boldsymbol{\mu}^{(i)})^\top \mathbf{J}_i \mathbf{E}_i \,\middle|\, \mathbf{I}_n \,\middle|\, \mathbf{0} \right] \in \mathbb{R}^{n \times p}.$$

For the observational covariance, vectorising $\boldsymbol{\Lambda}\boldsymbol{\Sigma} + \boldsymbol{\Sigma}\boldsymbol{\Lambda}^\top - \mathrm{diag}(\mathbf{d}) = 0$ and selecting the upper-triangular part yields

$$\mathbf{K}_0 = \left[ \mathbf{S}_n\big(\boldsymbol{\Sigma} \otimes \mathbf{I}_n + (\mathbf{I}_n \otimes \boldsymbol{\Sigma})\mathbf{C}\big) \,\middle|\, \mathbf{0} \,\middle|\, -\mathbf{S}_n\mathbf{P} \right] \in \mathbb{R}^{\frac{n(n+1)}{2} \times p}.$$

For the interventional covariance (on coordinate $i$), we define:

$$\mathbf{K}_1 = \left[ \mathbf{S}_n\big(\boldsymbol{\Sigma}^{(i)} \otimes \mathbf{I}_n + (\mathbf{I}_n \otimes \boldsymbol{\Sigma}^{(i)})\mathbf{C} - \big[(\mathbf{I}_n \otimes \mathbf{e}_i) + (\mathbf{e}_i \otimes \mathbf{I}_n)\big]\boldsymbol{\Sigma}^{(i)}\mathbf{J}_i\mathbf{E}_i\big) \,\middle|\, \mathbf{0} \,\middle|\, -\mathbf{S}_n\mathbf{P} \right] \in \mathbb{R}^{\frac{n(n+1)}{2} \times p}.$$

Assembling all the equations:

$$\mathbf{A} := \begin{bmatrix} \mathbf{M}_0 \\ \mathbf{M}_1 \\ \mathbf{K}_0 \\ \mathbf{K}_1 \end{bmatrix} \in \mathbb{R}^{m \times p}, \tag{13}$$

where $m = n + n + \frac{n(n+1)}{2} + \frac{n(n+1)}{2} = n^2 + 3n$.

## A.2. Notion of Genericity

Fix a directed graph $G$, and let $\Theta_G \subset \mathbb{R}^{|E|+3n}$ denote the admissible parameter space consisting of all free parameters $\theta = (\boldsymbol{\Lambda}, \mathbf{b}, \mathbf{d})$ compatible with $G$, with $\boldsymbol{\Lambda}$ positive stable and $\mathbf{D} = \mathrm{diag}(\mathbf{d}) \succ 0$. Here, compatibility with $G$ means that the off-diagonal support of $\boldsymbol{\Lambda}$ is a subset of $G$. The set $\Theta_G$ is open in $\mathbb{R}^{|E|+3n}$. We say that a property holds *generically* on $\Theta_G$ if there exists a Lebesgue-measure-zero subset $N_G \subset \Theta_G$ such that the property holds for every $\theta \in \Theta_G \setminus N_G$.

### A.3. Proof of Theorem 3.3

Throughout this proof, genericity is understood with respect to the admissible parameter space $\Theta_G$ for any given graph $G$, as defined in the genericity convention. The statement of this theorem only depends on $(\mathbf{\Lambda}, \mathbf{b})$, not on the diffusion parameters $\mathbf{d}$. Thus, any Lebesgue-measure-zero exceptional set in the free coordinates for $(\mathbf{\Lambda}, \mathbf{b})$ induces a Lebesgue-measure-zero exceptional set in the full admissible space $\Theta_G$. Moreover, by the model setup, the intervened matrix $\widetilde{\mathbf{\Lambda}}^{(i)}$ is invertible whenever the interventional mean $\boldsymbol{\mu}^{(i)}$ is considered.

At steady state, $\boldsymbol{\mu} = \mathbf{\Lambda}^{-1}\mathbf{b}$ and $\boldsymbol{\mu}^{(i)} = (\widetilde{\mathbf{\Lambda}}^{(i)})^{-1}\mathbf{b}$. Let $\mathbf{E} := \widetilde{\mathbf{\Lambda}}^{(i)} - \mathbf{\Lambda}$; only the $i$-th row of $\mathbf{E}$ is nonzero, with $E_{ij} = -\Lambda_{ij}$ for $j \neq i$. By the following equation,

$$\Delta\boldsymbol{\mu} := \boldsymbol{\mu} - \boldsymbol{\mu}^{(i)} = \mathbf{\Lambda}^{-1}\mathbf{E}\,(\widetilde{\mathbf{\Lambda}}^{(i)})^{-1}\mathbf{b} = \mathbf{\Lambda}^{-1}\mathbf{E}\,\boldsymbol{\mu}^{(i)}. \tag{14}$$

Since only row $i$ of $\mathbf{E}$ is nonzero, $\mathbf{E}\mathbf{v} = s(\mathbf{v})\,\mathbf{e}_i$ for any vector $\mathbf{v}$, where

$$s(\mathbf{v}) := -\sum_{j \neq i} \Lambda_{ij}\, v_j.$$

Applying this to $\mathbf{v} = \boldsymbol{\mu}^{(i)}$ in equation 14 gives

$$\Delta\boldsymbol{\mu} = s\big(\boldsymbol{\mu}^{(i)}\big)\,\mathbf{\Lambda}^{-1}\mathbf{e}_i. \tag{15}$$

Permute coordinates by some permutation matrix that orders the SCCs topologically. Let $\mathbf{L}$ be the block lower–triangular drift matrix after topologically ordering the SCCs,

$$\mathbf{L} = \begin{bmatrix} \mathbf{L}^{(1)} & 0 & \cdots & 0 \\ \mathbf{L}^{(2,1)} & \mathbf{L}^{(2)} & \ddots & \vdots \\ \vdots & \ddots & \ddots & 0 \\ \mathbf{L}^{(K,1)} & \cdots & \mathbf{L}^{(K,K-1)} & \mathbf{L}^{(K)} \end{bmatrix},$$

with each diagonal block $\mathbf{L}^{(u)}$ invertible. Fix a block $r$ and a coordinate $i$ in block $r$.

Since $\mathbf{L}$ is block lower–triangular, $\mathbf{L}^{-1}$ is also block lower–triangular. Therefore, $(\mathbf{L}^{-1}\mathbf{e}_i)_j = 0$ whenever the block containing $j$ is not a descendant of the block containing $i$. It remains to show that, for descendant coordinates, this entry is generically nonzero. We show that if $t$ is a descendant of $r$ in the DAG over SCCs and $j$ is a coordinate in block $t$, then there exists an admissible parameter value $\theta^\star \in \Theta_G$ such that $\big(\mathbf{L}(\theta^\star)^{-1}\mathbf{e}_i\big)_j \neq 0$. Indeed, by the adjugate formula, this entry can be written as $\big(\mathbf{L}(\theta)^{-1}\mathbf{e}_i\big)_j = \frac{p_{j,i}(\theta)}{\det \mathbf{L}(\theta)}$, where $p_{j,i}(\theta)$ is a polynomial in the free entries of $\mathbf{L}(\theta)$. Since $\det \mathbf{L}(\theta) \neq 0$ on the admissible parameter space, showing one admissible point where the entry is nonzero shows that $p_{j,i}$ is not identically zero. Hence the zero set of the entry is contained in the zero set of a nonzero polynomial, and therefore has Lebesgue measure zero.

Since $j$ lies in a descendant block of the block containing $i$, there exists a coordinate-level directed path in $G$ from $i$ to $j$, say $i = v_0 \to v_1 \to \cdots \to v_\ell = j$. If necessary, remove cycles so that the path is simple. We construct a one-parameter family $\mathbf{L}(\varepsilon)$ as follows. Choose $\alpha > 0$ sufficiently large. Set all diagonal entries of $\mathbf{L}(\varepsilon)$ equal to $\alpha$. For each edge $v_{q-1} \to v_q$ on the selected path, set the corresponding entry $L_{v_q,v_{q-1}}(\varepsilon)$ equal to 1. Set every other allowed off-diagonal entry of $\mathbf{L}(\varepsilon)$ equal to $\varepsilon$, and keep all forbidden entries equal to zero. For every $\varepsilon > 0$, every allowed off-diagonal entry is nonzero, so $\mathbf{L}(\varepsilon)$ has the exact support prescribed by $G$. Moreover, by choosing $\alpha$ large and $\varepsilon$ sufficiently small, $\mathbf{L}(\varepsilon)$ is strictly row-diagonally dominant with positive diagonal entries, hence positive stable and in particular invertible. Thus, after choosing arbitrary admissible $\mathbf{b}$ and any $\mathbf{d} \succ 0$, this construction gives a point $\theta(\varepsilon) \in \Theta_G$ for all sufficiently small $\varepsilon > 0$.

Let $\mathbf{P}_{\text{path}}$ be the matrix containing only the selected path entries, i.e., $(\mathbf{P}_{\text{path}})_{v_q,v_{q-1}} = 1$ for $q = 1, \ldots, \ell$ and all other entries zero. At $\varepsilon = 0$, $\mathbf{L}(0) = \alpha\mathbf{I} + \mathbf{P}_{\text{path}}$. Since the selected path is simple, $\mathbf{P}_{\text{path}}$ is nilpotent,[6] and

$$(\alpha\mathbf{I} + \mathbf{P}_{\text{path}})^{-1} = \alpha^{-1}\sum_{q \geq 0}(-\alpha^{-1}\mathbf{P}_{\text{path}})^q,$$

---

[6]A square matrix $\mathbf{A}$ is nilpotent if there exists an integer $m \geq 1$ such that $\mathbf{A}^m = \mathbf{0}$. In this proof, $\mathbf{P}_{\text{path}}$ is nilpotent because it only moves forward along the finite simple path $i = v_0 \to v_1 \to \cdots \to v_\ell = j$, so $\mathbf{P}_{\text{path}}^{\ell+1} = \mathbf{0}$.

where the sum is finite. Because $\mathbf{P}_{\text{path}}^{\ell}\mathbf{e}_i = \mathbf{e}_j$, while no lower power maps $\mathbf{e}_i$ to $\mathbf{e}_j$, we get

$$\left(\mathbf{L}(0)^{-1}\mathbf{e}_i\right)_j = (-1)^{\ell}\alpha^{-(\ell+1)} \neq 0.$$

By continuity, $\left(\mathbf{L}(\varepsilon)^{-1}\mathbf{e}_i\right)_j \neq 0$ for all sufficiently small $\varepsilon > 0$. Hence we have exhibited admissible parameters with $(\mathbf{L}^{-1}\mathbf{e}_i)_j \neq 0$.

It follows that, for each fixed descendant coordinate $j \in \text{Desc}(C)$, the exceptional set on which $(\mathbf{L}^{-1}\mathbf{e}_i)_j = 0$ has Lebesgue measure zero in $\Theta_G$. Since there are only finitely many coordinates, taking the union over all descendant coordinates still gives a Lebesgue-measure-zero exceptional set. Therefore, generically,

$$\left(\mathbf{L}^{-1}\mathbf{e}_i\right)_j \neq 0 \quad \text{for every } j \in \text{Desc}(C),$$

where $\text{Desc}(C)$ includes the component $C$ itself. On the other hand, as shown above, for $j \notin \text{Desc}(C)$, the block lower–triangular structure gives $\left(\mathbf{L}^{-1}\mathbf{e}_i\right)_j = 0$ deterministically.

If for some $j \neq i$, we have $\Lambda_{ij} \neq 0$, then $s(\boldsymbol{\mu}^{(i)}) = -\sum_{j\neq i}\Lambda_{ij}\mu_j^{(i)}$. This is an analytic function of $(\boldsymbol{\Lambda}, \mathbf{b})$. Moreover, if the $i$-th row is non-null, then the row vector $\mathbf{r}_i^{\top} := -\sum_{j\neq i}\Lambda_{ij}\mathbf{e}_j^{\top}$ is nonzero. Since $s(\boldsymbol{\mu}^{(i)}) = \mathbf{r}_i^{\top}(\widetilde{\boldsymbol{\Lambda}}^{(i)})^{-1}\mathbf{b}$, and $(\widetilde{\boldsymbol{\Lambda}}^{(i)})^{-1}$ is invertible, the row vector $\mathbf{r}_i^{\top}(\widetilde{\boldsymbol{\Lambda}}^{(i)})^{-1}$ is nonzero. Thus, for fixed $\boldsymbol{\Lambda}$, $s(\boldsymbol{\mu}^{(i)})$ is a nonzero linear function of $\mathbf{b}$. Hence it is not identically zero as a function of $(\boldsymbol{\Lambda}, \mathbf{b})$, and its zero set has Lebesgue measure zero in the admissible free-coordinate space. Therefore $s(\boldsymbol{\mu}^{(i)}) \neq 0$ generically.

Combining with equation 15, the support of $\Delta\boldsymbol{\mu}$ matches that of $\boldsymbol{\Lambda}^{-1}\mathbf{e}_i$ whenever $s(\boldsymbol{\mu}^{(i)}) \neq 0$. Thus, if for some $j \neq i$, $\Lambda_{ij} \neq 0$, generically,

$$\mu_k^{(i)} \neq \mu_k \iff k \in \text{Desc}(C),$$

where $\text{Desc}(C)$ includes $C$ itself.

If $\Lambda_{ij} = 0$ for all $j \neq i$, then $\mathbf{E} = \mathbf{0}$, so $s(\boldsymbol{\mu}^{(i)}) = 0$ and $\Delta\boldsymbol{\mu} = \mathbf{0}$. Hence $\boldsymbol{\mu}^{(i)} = \boldsymbol{\mu}$.

### A.4. Proof of Theorem 3.3 under a soft row intervention

We only indicate the changes relative to the proof of Theorem 3.3 for the hard intervention. The argument showing the generic support of $\boldsymbol{\Lambda}^{-1}\mathbf{e}_i$ is unchanged.

At steady state, $\boldsymbol{\mu} = \boldsymbol{\Lambda}^{-1}\mathbf{b}, \boldsymbol{\mu}^{(i)} = (\widetilde{\boldsymbol{\Lambda}}^{(i)})^{-1}\mathbf{b}$. For a soft row intervention, $\widetilde{\boldsymbol{\Lambda}}^{(i)}$ differs from $\boldsymbol{\Lambda}$ only in row $i$. The entries in row $i$, including the diagonal entry, may change. Let $\mathbf{E} := \widetilde{\boldsymbol{\Lambda}}^{(i)} - \boldsymbol{\Lambda}$. Then only row $i$ of $\mathbf{E}$ is nonzero. Denote this row change by $\boldsymbol{\delta}_i^{\top} := \mathbf{e}_i^{\top}\mathbf{E} = \mathbf{e}_i^{\top}(\widetilde{\boldsymbol{\Lambda}}^{(i)} - \boldsymbol{\Lambda})$. We assume the intervention is non-null, i.e., $\boldsymbol{\delta}_i \neq 0$.

As before,

$$\Delta\boldsymbol{\mu} := \boldsymbol{\mu} - \boldsymbol{\mu}^{(i)} = \boldsymbol{\Lambda}^{-1}\mathbf{E}(\widetilde{\boldsymbol{\Lambda}}^{(i)})^{-1}\mathbf{b} = \boldsymbol{\Lambda}^{-1}\mathbf{E}\boldsymbol{\mu}^{(i)}. \tag{16}$$

Since only row $i$ of $\mathbf{E}$ is nonzero, for any vector $\mathbf{v}$, $\mathbf{Ev} = \eta(\mathbf{v})\mathbf{e}_i$, where

$$\eta(\mathbf{v}) := \boldsymbol{\delta}_i^{\top}\mathbf{v} = \sum_{j=1}^{n}\left(\widetilde{\Lambda}_{ij}^{(i)} - \Lambda_{ij}\right)v_j.$$

Applying this to $\mathbf{v} = \boldsymbol{\mu}^{(i)}$ in equation 16 gives

$$\Delta\boldsymbol{\mu} = \eta(\boldsymbol{\mu}^{(i)})\boldsymbol{\Lambda}^{-1}\mathbf{e}_i. \tag{17}$$

The rest of the proof uses the same generic-support claim proved in the hard intervention case: for the SCC $C$ containing $i$,

$$(\boldsymbol{\Lambda}^{-1}\mathbf{e}_i)_k = 0 \quad \text{deterministically for } k \notin \text{Desc}(C),$$

and

$$(\boldsymbol{\Lambda}^{-1}\mathbf{e}_i)_k \neq 0 \quad \text{generically for every } k \in \text{Desc}(C),$$

where $\mathrm{Desc}(C)$ includes $C$ itself.

It remains only to check that the scalar factor in equation 17 is generically nonzero. We have $\eta(\boldsymbol{\mu}^{(i)}) = \boldsymbol{\delta}_i^\top (\widetilde{\boldsymbol{\Lambda}}^{(i)})^{-1} \mathbf{b}$. Since $\boldsymbol{\delta}_i \neq 0$ and $\widetilde{\boldsymbol{\Lambda}}^{(i)}$ is invertible, the row vector $\boldsymbol{\delta}_i^\top (\widetilde{\boldsymbol{\Lambda}}^{(i)})^{-1}$ is nonzero. Hence, for fixed $(\boldsymbol{\Lambda}, \widetilde{\boldsymbol{\Lambda}}^{(i)})$, $\eta(\boldsymbol{\mu}^{(i)})$ is a nonzero linear function of $\mathbf{b}$. Therefore its zero set is a Lebesgue-measure-zero exceptional set, and $\eta(\boldsymbol{\mu}^{(i)}) \neq 0$ generically.

Combining this with equation 17, the support of $\Delta\boldsymbol{\mu}$ generically agrees with the support of $\boldsymbol{\Lambda}^{-1}\mathbf{e}_i$. Therefore, for a non-null soft row intervention on coordinate $i$,

$$\mu_k^{(i)} \neq \mu_k \quad \Longleftrightarrow \quad k \in \mathrm{Desc}(C)$$

generically, where $\mathrm{Desc}(C)$ includes $C$ itself.

If the soft intervention is null, i.e., $\boldsymbol{\delta}_i = 0$, then $\mathbf{E} = 0$, so $\Delta\boldsymbol{\mu} = 0$ and $\boldsymbol{\mu}^{(i)} = \boldsymbol{\mu}$.

## A.5. Recovering SCCs and a topological order over SCCs from mean changes

Assume at least one intervention is performed in every SCC. For each intervention on $i$, set

$$R_i := \{\, j : \mu_j^{(i)} \neq \mu_j \,\}.$$

By Theorem 3.3, generically,

$$R_i = \mathrm{Desc}(C_i) \quad \text{if the intervention on } i \text{ is non-null,} \qquad R_i = \emptyset \quad \text{if it is null,}$$

where $C_i$ is the SCC containing $i$ and $\mathrm{Desc}(C)$ is the set of nodes lying in SCCs reachable from $C$ in the SCC–DAG (including $C$ itself).

**Procedure.**

1. *Singleton sources (null interventions).* If $R_i = \emptyset$, declare $\{i\}$ a singleton SCC with no incoming edges (a source). Do *not* merge different $i$ with $R_i = \emptyset$.

2. *Non-null SCCs.* For the remaining $i$, group by equality of sets: $i \sim i'$ iff $R_i = R_{i'}$. Each equivalence class is exactly one SCC; write $R_C$ for the common set of a class $C$.

3. *Edges among non-null SCCs.* There is an edge from $C$ to $C'$ if $R_C \supsetneq R_{C'}$ and no $D$ satisfying $R_C \supsetneq R_D \supsetneq R_{C'}$.

4. *Topological order.* Output all singleton sources from step 1 first (in any arbitrary order), then the non-null SCCs according to the recovered edges among non-null SCCs. This yields a valid topological ordering.

## A.6. Proof of Theorem 3.1

Throughout this proof, genericity is understood with respect to the admissible parameter space $\Theta_G$ for any fixed graph $G$, as defined in the genericity convention in Appendix A.2.

Consider the following two admissible triples satisfying the linear system equation 13:

- True underlying triple: $(\boldsymbol{\Lambda}_\star, \mathbf{b}_\star, \mathbf{D}_\star)$.

- Alternative admissible triple: $(\boldsymbol{\Lambda}', \mathbf{b}', \mathbf{D}')$.

Define

$$\Delta\boldsymbol{\Theta} := \begin{bmatrix} \mathrm{vec}(\Delta\boldsymbol{\Lambda}) \\ \Delta\mathbf{b} \\ \Delta\mathbf{d} \end{bmatrix}, \qquad \Delta\boldsymbol{\Lambda} := \boldsymbol{\Lambda}' - \boldsymbol{\Lambda}_\star, \quad \Delta\mathbf{b} := \mathbf{b}' - \mathbf{b}_\star, \quad \Delta\mathbf{D} := \mathbf{D}' - \mathbf{D}_\star,$$

where $\Delta\mathbf{d}$ is the diagonal of $\Delta\mathbf{D}$. Since $\mathbf{A}\boldsymbol{\Theta}_\star = 0$ and $\mathbf{A}\boldsymbol{\Theta}' = 0$, we have $\mathbf{A}\Delta\boldsymbol{\Theta} = 0$. Our goal is to show that the alternative triple is a scalar multiple of the true triple, i.e., $\boldsymbol{\Lambda}' = c\boldsymbol{\Lambda}_\star, \mathbf{b}' = c\mathbf{b}_\star, \mathbf{D}' = c\mathbf{D}_\star$ for some scalar $c > 0$.

Fix the intervened row index $i$. The observational blocks $\mathbf{M}_0$ and $\mathbf{K}_0$ yield

$$\Delta\mathbf{\Lambda}\boldsymbol{\mu} = \Delta\mathbf{b}, \qquad \Delta\mathbf{\Lambda}\mathbf{\Sigma} + \mathbf{\Sigma}\Delta\mathbf{\Lambda}^\top = \Delta\mathbf{D}, \tag{18}$$

where $\boldsymbol{\mu}$ and $\mathbf{\Sigma}$ are the true observational mean and covariance. Since the intervention zeros out row $i$ off-diagonals,

$$\widetilde{\mathbf{\Lambda}}_\star^{(i)} = \mathbf{J}_i \odot \mathbf{\Lambda}_\star, \qquad \widetilde{\mathbf{\Lambda}}'^{(i)} = \mathbf{J}_i \odot \mathbf{\Lambda}', \qquad \mathbf{J}_i := \mathbf{1}\mathbf{1}^\top - \mathbf{e}_i\mathbf{1}^\top + \mathbf{e}_i\mathbf{e}_i^\top.$$

Thus the interventional blocks $\mathbf{M}_1$ and $\mathbf{K}_1$ give

$$(\mathbf{J}_i \odot \Delta\mathbf{\Lambda})\boldsymbol{\mu}^{(i)} = \Delta\mathbf{b}, \qquad (\mathbf{J}_i \odot \Delta\mathbf{\Lambda})\mathbf{\Sigma}^{(i)} + \mathbf{\Sigma}^{(i)}(\Delta\mathbf{\Lambda}^\top \odot \mathbf{J}_i^\top) = \Delta\mathbf{D}. \tag{19}$$

Taking the $i$-th row of the first equation gives

$$(\Delta\mathbf{\Lambda})_{ii}\mu_i^{(i)} = \Delta b_i. \tag{20}$$

Outside the measure-zero exceptional set where $b_{\star i} = 0$, define $c := \frac{b_i'}{b_{\star i}}$. Define the scaled-difference variables

$$\mathbf{\Lambda}_\Delta := \mathbf{\Lambda}' - c\mathbf{\Lambda}_\star, \qquad \mathbf{b}_\Delta := \mathbf{b}' - c\mathbf{b}_\star, \qquad \mathbf{D}_\Delta := \mathbf{D}' - c\mathbf{D}_\star.$$

Then $(\mathbf{\Lambda}_\Delta, \mathbf{b}_\Delta, \mathbf{D}_\Delta)$ still satisfies the homogeneous linear system, and $(b_\Delta)_i = 0$. In what follows, all moments $\boldsymbol{\mu}, \mathbf{\Sigma}, \boldsymbol{\mu}^{(i)}, \mathbf{\Sigma}^{(i)}$ remain the true moments generated by $(\mathbf{\Lambda}_\star, \mathbf{b}_\star, \mathbf{D}_\star)$.

For the true intervened system, the $i$-th row gives $\Lambda_{\star,ii}\mu_i^{(i)} = b_{\star i}$. Since $b_{\star i} \neq 0$, we have $\mu_i^{(i)} \neq 0$. Applying equation 20 to the ambiguity triple and using $(b_\Delta)_i = 0$, we get $(\mathbf{\Lambda}_\Delta)_{ii} = 0$. Moreover, the $(i, i)$ entry of the interventional covariance equation gives $2\Sigma_{ii}^{(i)}(\mathbf{\Lambda}_\Delta)_{ii} = (\mathbf{D}_\Delta)_{ii}$, and hence $(\mathbf{D}_\Delta)_{ii} = 0$.

For the ambiguity triple, write

$$\widetilde{\mathbf{\Lambda}}_\Delta^{(i)} = \mathbf{J}_i \odot \mathbf{\Lambda}_\Delta = \begin{bmatrix} 0 & 0 \\ (\mathbf{\Lambda}_\Delta)_{-i,i} & (\mathbf{\Lambda}_\Delta)_{-i,-i} \end{bmatrix}.$$

Using the $(-i, i)$ block of the interventional covariance equation and the fact that $\mathbf{D}_\Delta$ is diagonal, we obtain

$$(\mathbf{\Lambda}_\Delta)_{-i,i}\Sigma_{ii}^{(i)} + (\mathbf{\Lambda}_\Delta)_{-i,-i}\mathbf{\Sigma}_{-i,i}^{(i)} = 0.$$

Thus

$$(\mathbf{\Lambda}_\Delta)_{-i,i} = -\frac{(\mathbf{\Lambda}_\Delta)_{-i,-i}\mathbf{\Sigma}_{-i,i}^{(i)}}{\Sigma_{ii}^{(i)}}. \tag{21}$$

The $(-i, -i)$ block then gives

$$(\mathbf{\Lambda}_\Delta)_{-i,-i}\mathbf{Z} + \mathbf{Z}(\mathbf{\Lambda}_\Delta)_{-i,-i}^\top = (\mathbf{D}_\Delta)_{-i,-i}, \tag{22}$$

where

$$\mathbf{Z} := \mathbf{\Sigma}_{-i,-i}^{(i)} - \frac{\mathbf{\Sigma}_{-i,i}^{(i)}\mathbf{\Sigma}_{i,-i}^{(i)}}{\Sigma_{ii}^{(i)}}.$$

The matrix $\mathbf{Z}$ is positive definite, since it is the Schur complement of $\Sigma_{ii}^{(i)}$ in $\mathbf{\Sigma}^{(i)} \succ 0$.

Next, the observational and interventional covariance equations for the ambiguity variables are

$$\mathbf{\Lambda}_\Delta\mathbf{\Sigma} + \mathbf{\Sigma}\mathbf{\Lambda}_\Delta^\top = \mathbf{D}_\Delta, \tag{23}$$
$$\widetilde{\mathbf{\Lambda}}_\Delta^{(i)}\mathbf{\Sigma}^{(i)} + \mathbf{\Sigma}^{(i)}\widetilde{\mathbf{\Lambda}}_\Delta^{(i)\top} = \mathbf{D}_\Delta. \tag{24}$$

Define $\mathbf{w}_\Delta \in \mathbb{R}^n$ by $(w_\Delta)_i = 0$ and $(\mathbf{w}_\Delta)_{-i} = (\mathbf{\Lambda}_\Delta)_{i,-i}^\top$, so that $\widetilde{\mathbf{\Lambda}}_\Delta^{(i)} = \mathbf{\Lambda}_\Delta - \mathbf{e}_i\mathbf{w}_\Delta^\top$. Subtracting equation 24 from equation 23, with $\mathbf{\Gamma} := \mathbf{\Sigma} - \mathbf{\Sigma}^{(i)}$, gives

$$\mathbf{\Lambda}_\Delta\mathbf{\Gamma} + \mathbf{\Gamma}\mathbf{\Lambda}_\Delta^\top + \mathbf{e}_i\mathbf{w}_\Delta^\top\mathbf{\Sigma}^{(i)} + \mathbf{\Sigma}^{(i)}\mathbf{w}_\Delta\mathbf{e}_i^\top = 0. \tag{25}$$

Substituting $\boldsymbol{\Lambda}_\Delta = \widetilde{\boldsymbol{\Lambda}}_\Delta^{(i)} + \mathbf{e}_i \mathbf{w}_\Delta^\top$ into equation 25, and then taking the $(-i, -i)$ block, removes all terms containing $\mathbf{e}_i$ and gives

$$\left( \widetilde{\boldsymbol{\Lambda}}_\Delta^{(i)} \boldsymbol{\Gamma} + \boldsymbol{\Gamma} \widetilde{\boldsymbol{\Lambda}}_\Delta^{(i)\top} \right)_{-i,-i} = 0. \tag{26}$$

Expanding this block,

$$(\boldsymbol{\Lambda}_\Delta)_{-i,i} \boldsymbol{\Gamma}_{i,-i} + (\boldsymbol{\Lambda}_\Delta)_{-i,-i} \boldsymbol{\Gamma}_{-i,-i} + \boldsymbol{\Gamma}_{-i,i} (\boldsymbol{\Lambda}_\Delta)_{-i,i}^\top + \boldsymbol{\Gamma}_{-i,-i} (\boldsymbol{\Lambda}_\Delta)_{-i,-i}^\top = 0. \tag{27}$$

Using equation 21, we obtain

$$(\boldsymbol{\Lambda}_\Delta)_{-i,-i} \boldsymbol{\Xi} + \boldsymbol{\Xi}^\top (\boldsymbol{\Lambda}_\Delta)_{-i,-i}^\top = 0, \tag{28}$$

where

$$\boldsymbol{\Xi} := \boldsymbol{\Gamma}_{-i,-i} - \frac{\boldsymbol{\Sigma}_{-i,i}^{(i)} \boldsymbol{\Gamma}_{i,-i}}{\boldsymbol{\Sigma}_{ii}^{(i)}}. \tag{29}$$

By Lemma A.3, $\boldsymbol{\Xi}$ is invertible. Define

$$\mathbf{A} := \boldsymbol{\Xi}^{-1} \mathbf{Z}, \qquad \mathbf{S} := (\boldsymbol{\Lambda}_\Delta)_{-i,-i} \boldsymbol{\Xi}.$$

Then equation 28 implies $\mathbf{S}^\top = -\mathbf{S}$. Moreover,

$$\mathbf{S}\mathbf{A} - \mathbf{A}^\top \mathbf{S} = (\boldsymbol{\Lambda}_\Delta)_{-i,-i} \mathbf{Z} + \mathbf{Z} (\boldsymbol{\Lambda}_\Delta)_{-i,-i}^\top = (\mathbf{D}_\Delta)_{-i,-i},$$

where the last equality follows from equation 22. Since $(\mathbf{D}_\Delta)_{-i,-i}$ is diagonal,

$$\mathrm{offdiag}(\mathbf{S}\mathbf{A} - \mathbf{A}^\top \mathbf{S}) = 0.$$

**Assumption A.1.** Let $\mathbf{A} := \boldsymbol{\Xi}^{-1} \mathbf{Z}$. Consider the linear map $\mathcal{L}_\mathbf{A} : \mathcal{K}_m \to \mathbb{R}_{\mathrm{off}}^{m \times m}$, where $\mathcal{K}_m := \{ \mathbf{S} \in \mathbb{R}^{m \times m} : \mathbf{S}^\top = -\mathbf{S} \}$, defined by

$$\mathcal{L}_\mathbf{A}(\mathbf{S}) = \mathrm{offdiag}(\mathbf{S}\mathbf{A} - \mathbf{A}^\top \mathbf{S}).$$

Let $\mathbf{L}_\mathbf{A}$ be the matrix representation of $\mathcal{L}_\mathbf{A}$ in any basis of $\mathcal{K}_m$. We assume $0 \notin \sigma(\mathbf{L}_\mathbf{A}^\top \mathbf{L}_\mathbf{A})$. Equivalently, the only skew-symmetric matrix $\mathbf{S}$ satisfying $\mathrm{offdiag}(\mathbf{S}\mathbf{A} - \mathbf{A}^\top \mathbf{S}) = 0$ is $\mathbf{S} = 0$.

By Assumption A.1, $\mathbf{S} = 0$. Since $\mathbf{S} = (\boldsymbol{\Lambda}_\Delta)_{-i,-i} \boldsymbol{\Xi}$ and $\boldsymbol{\Xi}$ is invertible, $(\boldsymbol{\Lambda}_\Delta)_{-i,-i} = 0$. Using equation 21, we also obtain $(\boldsymbol{\Lambda}_\Delta)_{-i,i} = 0$. Together with $(\boldsymbol{\Lambda}_\Delta)_{ii} = 0$, this shows that the only possibly nonzero entries of $\boldsymbol{\Lambda}_\Delta$ are the off-diagonal entries in row $i$. Moreover, from equation 22, we get $(\mathbf{D}_\Delta)_{-i,-i} = 0$. Since we already showed $(\mathbf{D}_\Delta)_{ii} = 0$, it follows that $\mathbf{D}_\Delta = 0$.

Now the observational covariance equation gives $\boldsymbol{\Lambda}_\Delta \boldsymbol{\Sigma} + \boldsymbol{\Sigma} \boldsymbol{\Lambda}_\Delta^\top = 0$. Because only row $i$ of $\boldsymbol{\Lambda}_\Delta$ can be nonzero, write $\boldsymbol{\Lambda}_\Delta = \mathbf{e}_i \mathbf{u}^\top$, $u_i = 0$. Then $\mathbf{e}_i \mathbf{u}^\top \boldsymbol{\Sigma} + \boldsymbol{\Sigma} \mathbf{u} \mathbf{e}_i^\top = 0$. Taking the $i$-th row gives $\mathbf{u}^\top \boldsymbol{\Sigma} + (\boldsymbol{\Sigma} \mathbf{u})_i \mathbf{e}_i^\top = 0$. For every $j \neq i$, this implies $(\mathbf{u}^\top \boldsymbol{\Sigma})_j = 0$, while the $(i, i)$ entry gives $2(\mathbf{u}^\top \boldsymbol{\Sigma})_i = 0$. Therefore $\mathbf{u}^\top \boldsymbol{\Sigma} = 0$. Since $\boldsymbol{\Sigma} \succ 0$ is invertible, $\mathbf{u} = 0$, and hence $\boldsymbol{\Lambda}_\Delta = 0$. Finally, the observational mean equation gives $\boldsymbol{\Lambda}_\Delta \boldsymbol{\mu} = \mathbf{b}_\Delta$, so $\mathbf{b}_\Delta = 0$. Hence the scaled-difference triple is zero: $\boldsymbol{\Lambda}' = c \boldsymbol{\Lambda}_\star, \mathbf{b}' = c \mathbf{b}_\star, \mathbf{D}' = c \mathbf{D}_\star$. Since $\mathbf{D}'$ and $\mathbf{D}_\star$ are positive diagonal matrices, we have $c > 0$. This completes the proof.

**Assumption A.2.** Consider right/left eigenbases of the true drift $\boldsymbol{\Lambda}_\star$: $\boldsymbol{\Lambda}_\star \mathbf{r}_{\star,\ell} = \lambda_{\star,\ell} \mathbf{r}_{\star,\ell}, \boldsymbol{\rho}_{\star,k}^\top \boldsymbol{\Lambda}_\star = \lambda_{\star,k} \boldsymbol{\rho}_{\star,k}^\top, \boldsymbol{\rho}_{\star,k}^\top \mathbf{r}_{\star,\ell} = \delta_{k\ell}$. Let $\mathbf{R}_\star := [\mathbf{r}_{\star,1} \cdots \mathbf{r}_{\star,n}], \mathbf{P}_\star := [\boldsymbol{\rho}_{\star,1} \cdots \boldsymbol{\rho}_{\star,n}]$, so that $\mathbf{P}_\star^\top \mathbf{R}_\star = \mathbf{I}$. Let $\mathbf{w}_\star \in \mathbb{R}^n$ be the true intervened-row vector, defined by $(w_\star)_i = 0$ and $(\mathbf{w}_\star)_{-i} = (\boldsymbol{\Lambda}_\star)_{i,-i}^\top$, so that $\widetilde{\boldsymbol{\Lambda}}_\star^{(i)} = \boldsymbol{\Lambda}_\star - \mathbf{e}_i \mathbf{w}_\star^\top$.

1. We assume that $\boldsymbol{\Lambda}_\star$ has simple spectrum, i.e., $\lambda_{\star,k} \neq \lambda_{\star,\ell}$ for $k \neq \ell$.

2. Let $\mathbf{W}_\star \in \mathbb{C}^{n \times n}$ be defined by

$$(\mathbf{W}_\star)_{k\ell} := \frac{(\boldsymbol{\rho}_{\star,k}^\top \mathbf{e}_i)(\mathbf{w}_\star^\top \boldsymbol{\Sigma}^{(i)} \boldsymbol{\rho}_{\star,\ell})}{\lambda_{\star,k} + \lambda_{\star,\ell}}.$$

We assume

$$0 \notin \sigma(\mathbf{W}_\star + \mathbf{W}_\star^\top).$$

3. With $\boldsymbol{\Gamma} := \boldsymbol{\Sigma} - \boldsymbol{\Sigma}^{(i)}$, we assume: $\mathbf{e}_i^\top \boldsymbol{\Gamma}^{-1} \boldsymbol{\Sigma}^{(i)} \mathbf{e}_i \neq 0$.

**Lemma A.3.** *Under Assumption A.2, the matrix $\boldsymbol{\Xi}$ is invertible.*

*Proof.* Since $\widetilde{\boldsymbol{\Lambda}}_\star^{(i)} = \boldsymbol{\Lambda}_\star - \mathbf{e}_i \mathbf{w}_\star^\top$, subtracting the true observational and interventional Lyapunov equations gives

$$\boldsymbol{\Lambda}_\star \boldsymbol{\Gamma} + \boldsymbol{\Gamma} \boldsymbol{\Lambda}_\star^\top + \mathbf{e}_i \mathbf{w}_\star^\top \boldsymbol{\Sigma}^{(i)} + \boldsymbol{\Sigma}^{(i)} \mathbf{w}_\star \mathbf{e}_i^\top = 0. \tag{30}$$

Let $\mathbf{U}$ solve the Sylvester equation

$$\boldsymbol{\Lambda}_\star \mathbf{U} + \mathbf{U} \boldsymbol{\Lambda}_\star^\top = -\mathbf{e}_i \big( \mathbf{w}_\star^\top \boldsymbol{\Sigma}^{(i)} \big). \tag{31}$$

Since $\boldsymbol{\Lambda}_\star$ is positive stable, the Sylvester operator is invertible. Transposing equation 31 and adding gives

$$\boldsymbol{\Lambda}_\star (\mathbf{U} + \mathbf{U}^\top) + (\mathbf{U} + \mathbf{U}^\top) \boldsymbol{\Lambda}_\star^\top = -\mathbf{e}_i \mathbf{w}_\star^\top \boldsymbol{\Sigma}^{(i)} - \boldsymbol{\Sigma}^{(i)} \mathbf{w}_\star \mathbf{e}_i^\top.$$

Comparing with equation 30 and using uniqueness gives

$$\boldsymbol{\Gamma} = \mathbf{U} + \mathbf{U}^\top. \tag{32}$$

Expand $\mathbf{U}$ in the right-eigenvector basis on both sides:

$$\mathbf{U} = \sum_{k,\ell} u_{k\ell} \, \mathbf{r}_{\star,k} \mathbf{r}_{\star,\ell}^\top,$$

where $u_{k\ell} = \boldsymbol{\rho}_{\star,k}^\top \mathbf{U} \boldsymbol{\rho}_{\star,\ell}$. Multiplying equation 31 on the left by $\boldsymbol{\rho}_{\star,k}^\top$ and on the right by $\boldsymbol{\rho}_{\star,\ell}$ gives

$$(\lambda_{\star,k} + \lambda_{\star,\ell}) u_{k\ell} = -(\boldsymbol{\rho}_{\star,k}^\top \mathbf{e}_i)\big( \mathbf{w}_\star^\top \boldsymbol{\Sigma}^{(i)} \boldsymbol{\rho}_{\star,\ell} \big).$$

Thus $u_{k\ell} = -(\mathbf{W}_\star)_{k\ell}$, and

$$\mathbf{U} = -\mathbf{R}_\star \mathbf{W}_\star \mathbf{R}_\star^\top. \tag{33}$$

Using equation 32,

$$\boldsymbol{\Gamma} = -\mathbf{R}_\star (\mathbf{W}_\star + \mathbf{W}_\star^\top) \mathbf{R}_\star^\top.$$

By Assumption A.2(2), $\mathbf{W}_\star + \mathbf{W}_\star^\top$ is invertible. Since $\mathbf{R}_\star$ is invertible, $\boldsymbol{\Gamma}$ is invertible.

Let $J = \{1, \ldots, n\} \setminus \{i\}$. Define

$$\mathbf{M}_i := \begin{bmatrix} \boldsymbol{\Sigma}_{ii}^{(i)} & \boldsymbol{\Gamma}_{i,J} \\ \boldsymbol{\Sigma}_{J,i}^{(i)} & \boldsymbol{\Gamma}_{J,J} \end{bmatrix}.$$

This matrix is obtained from $\boldsymbol{\Gamma}$ by replacing its $i$-th column with $\boldsymbol{\Sigma}^{(i)} \mathbf{e}_i$. Hence, by Cramer's rule,

$$\det(\mathbf{M}_i) = \det(\boldsymbol{\Gamma}) \, \mathbf{e}_i^\top \boldsymbol{\Gamma}^{-1} \boldsymbol{\Sigma}^{(i)} \mathbf{e}_i.$$

By Assumption A.2(3), $\det(\mathbf{M}_i) \neq 0$. On the other hand, by the Schur determinant identity,

$$\det(\mathbf{M}_i) = \boldsymbol{\Sigma}_{ii}^{(i)} \det(\boldsymbol{\Xi}).$$

Since $\boldsymbol{\Sigma}_{ii}^{(i)} > 0$, we get $\det(\boldsymbol{\Xi}) \neq 0$. Thus $\boldsymbol{\Xi}$ is invertible. $\qquad \square$

### A.7. Proof of Theorem 3.6

Let $\mathbf{T}$ be the target block. Suppose the parameters of block $\mathbf{P}$ satisfy $\mathbf{b_P} = c\,\bar{\mathbf{b}}_\mathbf{P}$ and $\mathbf{D_P} = c\,\bar{\mathbf{D}}_\mathbf{P}$ for a scalar $c > 0$ and known representatives $\bar{\mathbf{b}}_\mathbf{P}, \bar{\mathbf{D}}_\mathbf{P}$. Define the residual covariances and cross terms

$$\mathbf{Z} := \boldsymbol{\Sigma}_{\mathbf{T}|\mathbf{P}}, \quad \mathbf{Z}^{(i)} := \boldsymbol{\Sigma}_{\mathbf{T}|\mathbf{P}}^{(i)}, \quad \mathbf{B} := \boldsymbol{\Sigma}_{\mathbf{TP}} \boldsymbol{\Sigma}_{\mathbf{PP}}^{-1}, \quad \mathbf{B}^{(i)} := \boldsymbol{\Sigma}_{\mathbf{TP}}^{(i)} \big( \boldsymbol{\Sigma}_{\mathbf{PP}}^{(i)} \big)^{-1}.$$

Define the parent mean contribution $\mathbf{h_P} := \bar{\mathbf{b}}_{\mathbf{P}} - \bar{\mathbf{D}}_{\mathbf{P}}\boldsymbol{\Sigma}_{\mathbf{PP}}^{-1}\boldsymbol{\mu}_{\mathbf{P}}$. Then set

$$\mathbf{v} := \mathbf{B}\mathbf{h_P}, \quad \mathbf{v}^{(i)} := \mathbf{B}^{(i)}\mathbf{h_P}, \qquad \mathbf{Q} := \mathbf{B}\,\bar{\mathbf{D}}_{\mathbf{P}}\,\mathbf{B}^{\top}, \quad \mathbf{Q}^{(i)} := \mathbf{B}^{(i)}\bar{\mathbf{D}}_{\mathbf{P}}\,\mathbf{B}^{(i)\top}.$$

Here we used that the intervention is inside the target block $\mathbf{T}$, so the upstream parent moments $\boldsymbol{\mu}_{\mathbf{P}}$ and $\boldsymbol{\Sigma}_{\mathbf{PP}}$ are unchanged by the intervention. Indeed, under the hard intervention, row $i$ of the full drift has no off-diagonal entries. In particular, the $i$-th row of $\widetilde{\boldsymbol{\Lambda}}_{\mathbf{TP}}^{(i)}$ is zero and the $i$-th row of $\widetilde{\boldsymbol{\Lambda}}_{\mathbf{TT}}^{(i)}$ is $(\lambda_{ii}, 0)$. Taking the $i$-th row of the interventional cross-covariance Lyapunov block gives

$$\lambda_{ii}\boldsymbol{\Sigma}_{i,\mathbf{P}}^{(i)} + \boldsymbol{\Sigma}_{i,\mathbf{P}}^{(i)}\boldsymbol{\Lambda}_{\mathbf{PP}}^{\top} = 0.$$

Since $\lambda_{ii}\mathbf{I} + \boldsymbol{\Lambda}_{\mathbf{PP}}^{\top}$ is invertible, we obtain $\boldsymbol{\Sigma}_{i,\mathbf{P}}^{(i)} = 0$. Hence the $i$-th row of $\mathbf{B}^{(i)} = \boldsymbol{\Sigma}_{\mathbf{TP}}^{(i)}(\boldsymbol{\Sigma}_{\mathbf{PP}}^{(i)})^{-1}$ is zero, and therefore $(\mathbf{Q}^{(i)})_{i,-i} = (\mathbf{Q}^{(i)})_{-i,i} = 0$.

Under the intervention on coordinate $i$, the $i$-th row of $\mathbf{B}^{(i)}$ is zero, hence $(\mathbf{Q}^{(i)})_{i,-i} = 0$.

Partition the drift on $\mathbf{T}$ as

$$\boldsymbol{\Lambda}_{\mathbf{TT}} = \begin{bmatrix} \lambda_{ii} & \boldsymbol{\lambda}_{i,-i} \\ \boldsymbol{\lambda}_{-i,i} & \boldsymbol{\Lambda}_{-i,-i} \end{bmatrix}, \qquad \widetilde{\boldsymbol{\Lambda}}_{\mathbf{TT}}^{(i)} = \begin{bmatrix} \lambda_{ii} & 0 \\ \boldsymbol{\lambda}_{-i,i} & \boldsymbol{\Lambda}_{-i,-i} \end{bmatrix},$$

with unknown diagonal $\mathbf{D_T}$ and vector $\mathbf{b_T}$.

We first justify the residual mean equation. From the block mean equations

$$\boldsymbol{\Lambda}_{\mathbf{PP}}\boldsymbol{\mu}_{\mathbf{P}} = \mathbf{b_P}, \qquad \boldsymbol{\Lambda}_{\mathbf{TP}}\boldsymbol{\mu}_{\mathbf{P}} + \boldsymbol{\Lambda}_{\mathbf{TT}}\boldsymbol{\mu}_{\mathbf{T}} = \mathbf{b_T},$$

and from

$$\boldsymbol{\mu}_{\mathbf{T}} = \boldsymbol{\mu}_{\mathbf{T}|\mathbf{P}} + \mathbf{B}\boldsymbol{\mu}_{\mathbf{P}},$$

we obtain

$$\boldsymbol{\Lambda}_{\mathbf{TT}}\boldsymbol{\mu}_{\mathbf{T}|\mathbf{P}} = \mathbf{b_T} - (\boldsymbol{\Lambda}_{\mathbf{TP}} + \boldsymbol{\Lambda}_{\mathbf{TT}}\mathbf{B})\boldsymbol{\mu}_{\mathbf{P}}.$$

The cross-covariance block of the Lyapunov equation gives

$$\boldsymbol{\Lambda}_{\mathbf{TP}}\boldsymbol{\Sigma}_{\mathbf{PP}} + \boldsymbol{\Lambda}_{\mathbf{TT}}\boldsymbol{\Sigma}_{\mathbf{TP}} + \boldsymbol{\Sigma}_{\mathbf{TP}}\boldsymbol{\Lambda}_{\mathbf{PP}}^{\top} = 0.$$

Using $\mathbf{B} = \boldsymbol{\Sigma}_{\mathbf{TP}}\boldsymbol{\Sigma}_{\mathbf{PP}}^{-1}$, this implies

$$\boldsymbol{\Lambda}_{\mathbf{TP}} + \boldsymbol{\Lambda}_{\mathbf{TT}}\mathbf{B} = -\mathbf{B}\boldsymbol{\Sigma}_{\mathbf{PP}}\boldsymbol{\Lambda}_{\mathbf{PP}}^{\top}\boldsymbol{\Sigma}_{\mathbf{PP}}^{-1}.$$

The parent Lyapunov equation

$$\boldsymbol{\Lambda}_{\mathbf{PP}}\boldsymbol{\Sigma}_{\mathbf{PP}} + \boldsymbol{\Sigma}_{\mathbf{PP}}\boldsymbol{\Lambda}_{\mathbf{PP}}^{\top} = \mathbf{D_P}$$

then gives

$$\boldsymbol{\Lambda}_{\mathbf{TP}} + \boldsymbol{\Lambda}_{\mathbf{TT}}\mathbf{B} = \mathbf{B}\boldsymbol{\Lambda}_{\mathbf{PP}} - \mathbf{B}\mathbf{D_P}\boldsymbol{\Sigma}_{\mathbf{PP}}^{-1}.$$

Since $\mathbf{b_P} = c\bar{\mathbf{b}}_{\mathbf{P}}$ and $\mathbf{D_P} = c\bar{\mathbf{D}}_{\mathbf{P}}$, we get

$$\boldsymbol{\Lambda}_{\mathbf{TT}}\boldsymbol{\mu}_{\mathbf{T}|\mathbf{P}} = \mathbf{b_T} - c\,\mathbf{B}\left(\bar{\mathbf{b}}_{\mathbf{P}} - \bar{\mathbf{D}}_{\mathbf{P}}\boldsymbol{\Sigma}_{\mathbf{PP}}^{-1}\boldsymbol{\mu}_{\mathbf{P}}\right) = \mathbf{b_T} - c\mathbf{v}.$$

On $\mathbf{T}$, the stationary equations are

$$\boldsymbol{\Lambda}_{\mathbf{TT}}\mathbf{Z} + \mathbf{Z}\boldsymbol{\Lambda}_{\mathbf{TT}}^{\top} = \mathbf{D_T} + c\,\mathbf{Q}, \qquad \boldsymbol{\Lambda}_{\mathbf{TT}}\boldsymbol{\mu}_{\mathbf{T}|\mathbf{P}} = \mathbf{b_T} - c\,\mathbf{v},$$

and, under intervention,

$$\widetilde{\boldsymbol{\Lambda}}_{\mathbf{TT}}^{(i)}\mathbf{Z}^{(i)} + \mathbf{Z}^{(i)}\widetilde{\boldsymbol{\Lambda}}_{\mathbf{TT}}^{(i)\top} = \mathbf{D_T} + c\,\mathbf{Q}^{(i)}, \qquad \widetilde{\boldsymbol{\Lambda}}_{\mathbf{TT}}^{(i)}\boldsymbol{\mu}_{\mathbf{T}|\mathbf{P}}^{(i)} = \mathbf{b_T} - c\,\mathbf{v}^{(i)}. \tag{34}$$

Subtracting gives the difference relations

$$\boldsymbol{\Lambda}_{\mathbf{TT}}\mathbf{Z} + \mathbf{Z}\boldsymbol{\Lambda}_{\mathbf{TT}}^{\top} - \left(\widetilde{\boldsymbol{\Lambda}}_{\mathbf{TT}}^{(i)}\mathbf{Z}^{(i)} + \mathbf{Z}^{(i)}\widetilde{\boldsymbol{\Lambda}}_{\mathbf{TT}}^{(i)\top}\right) = c\,(\mathbf{Q} - \mathbf{Q}^{(i)}),$$

$$\boldsymbol{\Lambda}_{\mathbf{TT}}\boldsymbol{\mu}_{\mathbf{T}|\mathbf{P}} - \widetilde{\boldsymbol{\Lambda}}_{\mathbf{TT}}^{(i)}\boldsymbol{\mu}_{\mathbf{T}|\mathbf{P}}^{(i)} = c\,(\mathbf{v}^{(i)} - \mathbf{v}).$$

**Interventional** $(-i, i)$ **block.** Taking the $(-i, i)$ block of the interventional Lyapunov equation and using that $\mathbf{D_T}$ is diagonal and $(\mathbf{Q}^{(i)})_{-i,i} = 0$ yields

$$\lambda_{ii}\, \mathbf{Z}^{(i)}_{-i,i} + \boldsymbol{\lambda}_{-i,i}\, Z^{(i)}_{ii} + \boldsymbol{\Lambda}_{-i,-i}\, \mathbf{Z}^{(i)}_{-i,i} = 0.$$

Since $Z^{(i)}_{ii} > 0$, write

$$\boldsymbol{u} := \frac{\mathbf{Z}^{(i)}_{-i,i}}{Z^{(i)}_{ii}}$$

to obtain

$$\boldsymbol{\lambda}_{-i,i} + (\boldsymbol{\Lambda}_{-i,-i} + \lambda_{ii}\mathbf{I})\,\boldsymbol{u} = 0. \tag{35}$$

**The $\boldsymbol{\Xi}$-equation on** $(-i, -i)$. Let $\boldsymbol{\Gamma} := \mathbf{Z} - \mathbf{Z}^{(i)}$ and define the data-only matrix

$$\boldsymbol{\Xi} := \boldsymbol{\Gamma}_{-i,-i} - \boldsymbol{u}\,\boldsymbol{\Gamma}_{i,-i}.$$

From the $(-i, -i)$ block of the difference Lyapunov equation and the identity above,

$$\boldsymbol{\Lambda}_{-i,-i}\,\boldsymbol{\Xi} + \boldsymbol{\Xi}^\top \boldsymbol{\Lambda}^\top_{-i,-i} = c\,(\mathbf{Q} - \mathbf{Q}^{(i)})_{-i,-i} + \lambda_{ii}\left(\boldsymbol{u}\,\boldsymbol{\Gamma}_{i,-i} + \boldsymbol{\Gamma}_{-i,i}\,\boldsymbol{u}^\top\right). \tag{36}$$

From equation 34 and equation 35, we derive

$$\mathrm{offdiag}\!\left(\boldsymbol{\Lambda}_{-i,-i}\,\mathbf{S}^{(i)} + \mathbf{S}^{(i)}\,\boldsymbol{\Lambda}^\top_{-i,-i}\right) - \lambda_{ii}\,\mathrm{offdiag}\!\left(\boldsymbol{u}\,\mathbf{Z}^{(i)}_{i,-i} + \mathbf{Z}^{(i)}_{-i,i}\,\boldsymbol{u}^\top\right) = \mathrm{offdiag}\!\left(c\,\mathbf{Q}^{(i)}_{-i,-i}\right), \tag{37}$$

with

$$\mathbf{S}^{(i)} := \mathbf{Z}^{(i)}_{-i,-i} - \boldsymbol{u}\,\mathbf{Z}^{(i)}_{i,-i} \succ 0.$$

Define the linear maps

$$\mathcal{A}_S(\mathbf{X}) := \mathrm{offdiag}(\mathbf{X}\mathbf{S}^{(i)} + \mathbf{S}^{(i)}\mathbf{X}^\top), \qquad \mathcal{A}_\Xi(\mathbf{X}) := \mathbf{X}\boldsymbol{\Xi} + \boldsymbol{\Xi}^\top\mathbf{X}^\top.$$

Equations equation 36 and equation 37 form a stacked linear system in $\boldsymbol{\Lambda}_{-i,-i}$,

$$\begin{bmatrix}\mathcal{A}_S\\\mathcal{A}_\Xi\end{bmatrix}(\boldsymbol{\Lambda}_{-i,-i}) = c\begin{bmatrix}\mathrm{offdiag}(\mathbf{Q}^{(i)}_{-i,-i})\\(\mathbf{Q} - \mathbf{Q}^{(i)})_{-i,-i}\end{bmatrix} + \lambda_{ii}\begin{bmatrix}\mathrm{offdiag}\!\left(\boldsymbol{u}\,\mathbf{Z}^{(i)}_{i,-i} + \mathbf{Z}^{(i)}_{-i,i}\,\boldsymbol{u}^\top\right)\\\boldsymbol{u}\,\boldsymbol{\Gamma}_{i,-i} + \boldsymbol{\Gamma}_{-i,i}\,\boldsymbol{u}^\top\end{bmatrix}.$$

Let $\mathcal{V}_G$ be the linear space of matrices with the allowed sparsity pattern of $\boldsymbol{\Lambda}_{-i,-i}$. Define

$$\mathcal{L}_{\mathrm{blk}}(\mathbf{X}) := \begin{bmatrix}\mathrm{offdiag}(\mathbf{X}\mathbf{S}^{(i)} + \mathbf{S}^{(i)}\mathbf{X}^\top)\\\mathbf{X}\boldsymbol{\Xi} + \boldsymbol{\Xi}^\top\mathbf{X}^\top\end{bmatrix}, \qquad \mathbf{X} \in \mathcal{V}_G.$$

Let $\mathbf{L}_{\mathrm{blk}}$ be any matrix representation of $\mathcal{L}_{\mathrm{blk}}$. At the point $\boldsymbol{\Lambda}_{\mathbf{TP}} = 0$, the residual target-block equations reduce to the single-SCC equations. Hence, under the single-SCC nondegeneracy assumptions, the stacked operator $\mathcal{L}_{\mathrm{blk}}$ is injective at this point. Since the entries of a matrix representation of $\mathcal{L}_{\mathrm{blk}}$ depend continuously, indeed rationally, on the model parameters, this full-column-rank condition is not identically violated and therefore holds generically. Thus, generically, the stacked linear system uniquely determines $\boldsymbol{\Lambda}_{-i,-i}$ from its right-hand side. Since the right-hand side is affine in $c$ and $\lambda_{ii}$, there exist data-dependent matrices $\mathbf{K}_0, \mathbf{K}_2$ such that

$$\boldsymbol{\Lambda}_{-i,-i} = c\,\mathbf{K}_0 + \lambda_{ii}\,\mathbf{K}_2. \tag{38}$$

From equation 35,

$$\boldsymbol{\lambda}_{-i,i} = -\big(c\mathbf{K}_0 + \lambda_{ii}\mathbf{K}_2 + \lambda_{ii}\mathbf{I}\big)\boldsymbol{u}.$$

Using the observational $(i, -i)$ block of the Lyapunov equation,

$$\lambda_{ii}\mathbf{Z}_{i,-i} + \boldsymbol{\lambda}_{i,-i}\mathbf{Z}_{-i,-i} + Z_{ii}\boldsymbol{\lambda}^\top_{-i,i} + \mathbf{Z}_{i,-i}\boldsymbol{\Lambda}^\top_{-i,-i} = c\mathbf{Q}_{i,-i}.$$

Since $\mathbf{Z}_{-i,-i} \succ 0$, we obtain

$$\boldsymbol{\lambda}_{i,-i} = \left( c\mathbf{Q}_{i,-i} - \lambda_{ii}\mathbf{Z}_{i,-i} - Z_{ii}\boldsymbol{\lambda}^{\top}_{-i,i} - \mathbf{Z}_{i,-i}\boldsymbol{\Lambda}^{\top}_{-i,-i} \right)\mathbf{Z}^{-1}_{-i,-i}.$$

Substituting the expressions for $\boldsymbol{\Lambda}_{-i,-i}$ and $\boldsymbol{\lambda}_{-i,i}$ gives

$$\boldsymbol{\lambda}_{i,-i} = \mathbf{A}_0 + \lambda_{ii}\mathbf{A}_1, \tag{39}$$

where

$$\mathbf{A}_0 = c\left( \mathbf{Q}_{i,-i} + Z_{ii}\boldsymbol{u}^{\top}\mathbf{K}_0^{\top} - \mathbf{Z}_{i,-i}\mathbf{K}_0^{\top} \right)\mathbf{Z}^{-1}_{-i,-i},$$

and

$$\mathbf{A}_1 = \left( -\mathbf{Z}_{i,-i} + Z_{ii}\boldsymbol{u}^{\top}(\mathbf{I} + \mathbf{K}_2^{\top}) - \mathbf{Z}_{i,-i}\mathbf{K}_2^{\top} \right)\mathbf{Z}^{-1}_{-i,-i}.$$

Thus $\mathbf{A}_0$ scales linearly with $c$, while $\mathbf{K}_2$ and $\mathbf{A}_1$ are data-dependent and do not scale with $c$.

From the $i$-th component of the mean difference, let

$$\Delta\mu_i := (\boldsymbol{\mu}_{\mathbf{T}|\mathbf{P}})_i - (\boldsymbol{\mu}^{(i)}_{\mathbf{T}|\mathbf{P}})_i, \qquad \Delta v_i := v_i^{(i)} - v_i.$$

Then

$$\lambda_{ii}\,\Delta\mu_i + \boldsymbol{\lambda}_{i,-i}(\boldsymbol{\mu}_{\mathbf{T}|\mathbf{P}})_{-i} = c\,\Delta v_i.$$

Insert $\boldsymbol{\lambda}_{i,-i} = \mathbf{A}_0 + \lambda_{ii}\mathbf{A}_1$ and group the terms to obtain

$$\lambda_{ii}\left(\Delta\mu_i + \mathbf{A}_1(\boldsymbol{\mu}_{\mathbf{T}|\mathbf{P}})_{-i}\right) = c\Delta v_i - \mathbf{A}_0(\boldsymbol{\mu}_{\mathbf{T}|\mathbf{P}})_{-i}.$$

Hence $\lambda_{ii}$ is identified as

$$\lambda_{ii} = \frac{c\Delta v_i - \mathbf{A}_0(\boldsymbol{\mu}_{\mathbf{T}|\mathbf{P}})_{-i}}{\Delta\mu_i + \mathbf{A}_1(\boldsymbol{\mu}_{\mathbf{T}|\mathbf{P}})_{-i}},$$

where it is determined up to the same scale $c$.

With $\lambda_{ii}$ identified, the previous equations give

$$\boldsymbol{\Lambda}_{-i,-i} = c\mathbf{K}_0 + \lambda_{ii}\,\mathbf{K}_2, \qquad \boldsymbol{\lambda}_{-i,i} = -\left( c\mathbf{K}_0 + \lambda_{ii}\,\mathbf{K}_2 + \lambda_{ii}\mathbf{I} \right)\boldsymbol{u}, \qquad \boldsymbol{\lambda}_{i,-i} = \mathbf{A}_0 + \lambda_{ii}\mathbf{A}_1.$$

Thus the entire $\boldsymbol{\Lambda}_{\mathbf{TT}}$ is determined with the same scaling $c$.

Finally,

$$\mathbf{D}_{\mathbf{T}} = \mathrm{diag}\left(\boldsymbol{\Lambda}_{\mathbf{TT}}\mathbf{Z} + \mathbf{Z}\boldsymbol{\Lambda}^{\top}_{\mathbf{TT}} - c\,\mathbf{Q}\right),$$

and

$$\mathbf{b}_{\mathbf{T}} = \boldsymbol{\Lambda}_{\mathbf{TT}}\boldsymbol{\mu}_{\mathbf{T}|\mathbf{P}} + c\,\mathbf{v}.$$

Thus $(\boldsymbol{\Lambda}_{\mathbf{TT}}, \mathbf{D}_{\mathbf{T}}, \mathbf{b}_{\mathbf{T}})$ are identified up to the same multiplicative scale $c$.

### A.8. An Example with Multiple Root SCCs

Consider an OU process with state indices $1, 2, 3, 4$, where $1, 2, 3$ are root singletons and $4$ is a child singleton. Let

$$\boldsymbol{\Lambda} = \begin{bmatrix} \lambda_1 & 0 & 0 & 0 \\ 0 & \lambda_2 & 0 & 0 \\ 0 & 0 & \lambda_3 & 0 \\ a_1 & a_2 & a_3 & \lambda_4 \end{bmatrix}, \qquad \mathbf{D} = \mathrm{diag}(d_1, d_2, d_3, d_4),$$

and assume $\mathbf{D}$ is diagonal and each $\lambda_i > 0$. Suppose we observe the steady-state mean and covariance $(\boldsymbol{\mu}, \boldsymbol{\Sigma})$ and also the interventional moments $(\boldsymbol{\mu}^{(4)}, \boldsymbol{\Sigma}^{(4)})$ under an intervention on node 4 that zeros the entries $a_i$s (please note that interventions on root give the same mean and covariance as the observational ones).

For each root $i \in \{1, 2, 3\}$, the scalar OU gives

$$\mu_i = \frac{b_i}{\lambda_i}, \qquad s_i := \Sigma_{ii} = \frac{d_i}{2\lambda_i}.$$

Let $t_i := \Sigma_{i4}$ denote the observed cross-covariances with the child. The off-diagonal Lyapunov equations yield

$$(\lambda_i + \lambda_4)t_i + s_i a_i = 0 \quad \Longleftrightarrow \quad a_i = -\frac{\lambda_i + \lambda_4}{s_i} t_i. \tag{40}$$

The child's observed mean and variance satisfy

$$\lambda_4 \mu_4 + \sum_{i=1}^{3} a_i \mu_i = b_4, \tag{41}$$

$$2\lambda_4 s_4 + 2 \sum_{i=1}^{3} a_i t_i = d_4. \tag{42}$$

Under the intervention on $4$ (which zeros the $a_i$s), we also observe

$$\mu_4^{(4)} = \frac{b_4}{\lambda_4}, \qquad 2\lambda_4 s_4^{(4)} = d_4, \qquad t_i^{(4)} = 0. \tag{43}$$

Now, fix any positive $(u_1, u_2, u_3)$ and a positive $v$ (to be chosen). Define

$$\lambda_i' = u_i \lambda_i, \quad b_i' = u_i b_i, \quad d_i' = u_i d_i \ (i = 1, 2, 3), \qquad \lambda_4' = v\lambda_4, \quad b_4' = v b_4, \quad d_4' = v d_4,$$

and choose $a_i'$ to preserve the observed $t_i$:

$$(\lambda_i' + \lambda_4')t_i + s_i a_i' = 0 \quad \Longleftrightarrow \quad a_i' = -\frac{u_i \lambda_i + v\lambda_4}{s_i} t_i. \tag{44}$$

Then for each root, $\mu_i' = \frac{b_i'}{\lambda_i'} = \mu_i$ and $s_i' = \frac{d_i'}{2\lambda_i'} = s_i$, while equation 44 ensures the same $t_i$.

For the interventional moments on 4, scaling $(\lambda_4, b_4, d_4)$ by the common factor $v$ preserves $\mu_4^{(4)} = \frac{b_4}{\lambda_4}$ and $s_4^{(4)} = \frac{d_4}{2\lambda_4}$, and $t_i^{(4)} = 0$ still holds since the $a_i$ are zeroed under intervention. Thus all interventional moments match equation 43.

It remains to enforce that the primed parameters also satisfy the child's *observational* equations equation 41–equation 42. Subtracting $v$ times the original equation 42 from the primed version gives

$$2\lambda_4' s_4 - 2v\lambda_4 s_4 + 2 \sum_{i=1}^{3} (a_i' - v a_i)t_i = 0 \quad \Longleftrightarrow \quad \sum_{i=1}^{3} (a_i' - v a_i)t_i = 0.$$

Using equation 40 and equation 44,

$$a_i' - v a_i = -\frac{u_i \lambda_i + v\lambda_4}{s_i} t_i + v\frac{\lambda_i + \lambda_4}{s_i} t_i = \frac{(v - u_i)\lambda_i}{s_i} t_i.$$

Hence

$$\sum_{i=1}^{3} (v - u_i)\lambda_i \frac{t_i^2}{s_i} = 0 \quad \Longleftrightarrow \quad v \sum_{i=1}^{3} \alpha_i = \sum_{i=1}^{3} u_i \alpha_i, \qquad \alpha_i := \lambda_i \frac{t_i^2}{s_i} > 0. \tag{45}$$

Thus

$$v = \frac{\sum_i u_i \alpha_i}{\sum_i \alpha_i}. \tag{46}$$

Similarly, subtracting $v$ times equation 41 from the primed version gives

$$\lambda_4' \mu_4 - v\lambda_4 \mu_4 + \sum_{i=1}^{3} (a_i' - v a_i)\mu_i = 0 \quad \Longleftrightarrow \quad \sum_{i=1}^{3} (v - u_i)\lambda_i \frac{t_i}{s_i}\mu_i = 0,$$

i.e.,

$$v \sum_{i=1}^{3} \beta_i = \sum_{i=1}^{3} u_i \beta_i, \qquad \beta_i := \lambda_i \frac{t_i}{s_i} \mu_i = \frac{b_i t_i}{s_i}. \tag{47}$$

Thus also

$$v = \frac{\sum_i u_i \beta_i}{\sum_i \beta_i}. \tag{48}$$

Equating equation 46 and equation 48 yields a single linear constraint on $(u_1, u_2, u_3)$:

$$\frac{\sum_i u_i \alpha_i}{\sum_i \alpha_i} = \frac{\sum_i u_i \beta_i}{\sum_i \beta_i} \iff \sum_{i=1}^{3} u_i \Big( \alpha_i \sum_j \beta_j - \beta_i \sum_j \alpha_j \Big) = 0. \tag{49}$$

Let $\gamma_i := \alpha_i \sum_j \beta_j - \beta_i \sum_j \alpha_j$. Unless $(\gamma_1, \gamma_2, \gamma_3) = (0, 0, 0)$, equation 49 defines a nontrivial hyperplane in $\mathbb{R}^3$ (Note that the condition $\gamma_i = 0$ for all $i$ is nongeneric. Indeed, it requires $\frac{\beta_1}{\alpha_1} = \frac{\beta_2}{\alpha_2} = \frac{\beta_3}{\alpha_3}$, i.e., $\frac{b_i}{\lambda_i t_i}$ is constant across $i$. Moreover, $\sum_i \gamma_i = 0$, so the hyperplane contains $(1, 1, 1)$. Therefore it contains positive choices of $(u_1, u_2, u_3)$ arbitrarily close to $(1, 1, 1)$, including choices that are not all equal. This imposes two independent algebraic constraints on the continuous parameters $(b_i, \lambda_i, t_i)$, hence holds only on a measure-zero subset of the parameter space. Therefore, the set of positive $(u_1, u_2, u_3)$ satisfying equation 49 has (at least) two degrees of freedom. For any such $(u_1, u_2, u_3)$, take $v$ from equation 46 (which equals equation 48) and define $a'_i$ by equation 44. Unless $u_1 = u_2 = u_3 = v$, the transformation is not a global rescaling of $(\mathbf{\Lambda}, \mathbf{b}, \mathbf{D})$, yet all moments (observational and interventional under node-4 intervention) coincide. Hence, identifiability up to a single global scale fails.

## B. Details of Experiments

### B.1. Synthetic Data

We generate synthetic datasets from a stable OU model. For each setup $|\mathcal{I}| \in \{0, 2, 4, 6, 8, 10\}$ (number of single-node interventions) and 40 instances, we fix $n = 10$ variables and sample:

- **Drift matrix $\mathbf{\Lambda}$**: start from zeros; for $i \neq j$, set $\Lambda_{ij} \sim \mathcal{N}(0.2, \sigma^2)$ with probability $\rho$ and 0 otherwise (default $\rho = 0.3$, $\sigma = 0.8$). Enforce positive stability by making each row strictly diagonally dominant:

$$\Lambda_{ii} \leftarrow \max\Big(\texttt{DIAG\_MIN}, \sum_j |\Lambda_{ij}| + 0.2 + u_i\Big), \quad u_i \sim \text{Unif}[0, 0.3], \ \texttt{DIAG\_MIN} = 0.8.$$

- **Bias vector and diffusion matrix**: $\mathbf{b} \sim \text{Unif}[0.2, 1.5]^n$; $\mathbf{D} = \text{diag}(\mathbf{d})$ with $d_k \sim \text{Unif}[0.2, 0.4]$.

Observational moments are computed via $\boldsymbol{\mu} = \mathbf{\Lambda}^{-1}\mathbf{b}$ and $\mathbf{\Lambda}\mathbf{\Sigma} + \mathbf{\Sigma}\mathbf{\Lambda}^\top = \mathbf{D}$ (continuous-time Lyapunov). An intervention on node $k$ is implemented by zeroing row $k$ of $\mathbf{\Lambda}$ and restoring its diagonal entry (self-regulation preserved), leaving $\mathbf{b}$ and $\mathbf{D}$ unchanged; intervened moments $(\boldsymbol{\mu}^{(k)}, \mathbf{\Sigma}^{(k)})$ are computed analogously. We then draw snapshots from the corresponding Gaussians (by default 40,000 observational samples and 20,000 samples per intervention).

### B.2. Real Data

**Library-size normalization.** For each cell $k \in \{1, \ldots, T_c\}$ in context/intervention $c$ with raw counts $\{x_k^i\}_{i=1}^N$ (genes $i = 1, \ldots, N$), define the library size $s_k = \sum_{i=1}^N x_k^i$. We normalize by fractions $f_k^i = x_k^i / s_k$ and then scale to a common size

$$\tilde{x}_k^i \ = \ 10^4 f_k^i \ = \ 10^4 \frac{x_k^i}{s_k}.$$

All downstream statistics are computed on $\tilde{x}_k^i$. We denote the cell vector $\tilde{\mathbf{x}}_k = (\tilde{x}_k^1, \ldots, \tilde{x}_k^N)^\top \in \mathbb{R}^N$.

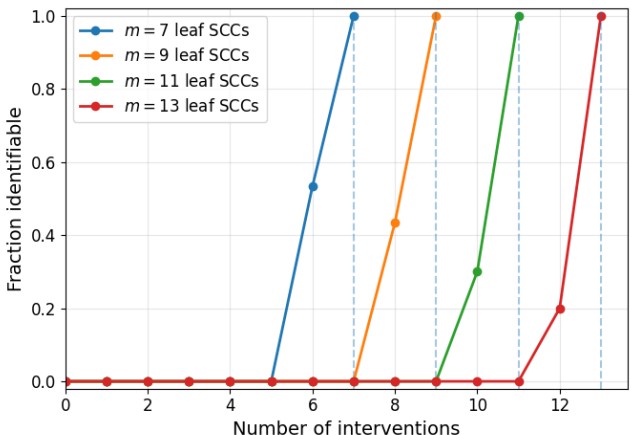

*Figure 2.* Identifiability versus number of interventions in the star condensation DAG.

**Empirical moments with shrinkage.** Given a context $c$ with $T_c$ cells $\{\tilde{\mathbf{x}}_k\}_{k=1}^{T_c}$, the empirical mean and (shrunk) covariance are

$$\hat{\boldsymbol{\mu}}_c = \frac{1}{T_c} \sum_{k=1}^{T_c} \tilde{\mathbf{x}}_k, \qquad \hat{\boldsymbol{\Sigma}}_c = (1 - \eta_c)\, \hat{\boldsymbol{\Sigma}}_{c,\text{sample}} + \eta_c\, \text{diag}\big(\hat{\boldsymbol{\Sigma}}_{c,\text{sample}}\big),$$

where $\hat{\boldsymbol{\Sigma}}_{c,\text{sample}} = \frac{1}{T_c - 1} \sum_{k=1}^{T_c} (\tilde{\mathbf{x}}_k - \hat{\boldsymbol{\mu}}_c)(\tilde{\mathbf{x}}_k - \hat{\boldsymbol{\mu}}_c)^\top$ and $\eta_c = \min\big(0.3,\ 10/(T_c - 1)\big)$.

**Model.** We fit an OU model with parameters $(\boldsymbol{\Lambda}, \mathbf{b}, \boldsymbol{D})$ on wild-type and interventional data. For an intervention on gene $k$, the context-specific drift is obtained from $\boldsymbol{\Lambda}$ by zeroing all off-diagonal entries in row $k$ and keeping $\Lambda_{kk}$; $\mathbf{b}$ and $\boldsymbol{D}$ are shared. We enforce $\text{diag}(\boldsymbol{\Lambda}) > 0$ (softplus) and rescale to $\text{tr}(\boldsymbol{\Lambda}) = n$.

**Train/test split and metrics.** Similar to previous work (Rohbeck et al., 2024), interventions are split into train context IDs $\{0, \ldots, 54\}$ and held-out test context IDs $\{55, \ldots, 60\}$. Evaluation uses the same per-cell normalization on two metric DES and PDS.

### B.3. Empirical study of graphical conditions

We emphasize that our identifiability result provides *sufficient* conditions. Thus, there may exist linear SDEs whose parameters are identifiable up to a global scaling even if they do not satisfy the graphical conditions in Theorem 3.6. To empirically study the conservativeness of the sufficient conditions in Theorem 3.6, we constructed a family of OU models whose condensation graph forms a star over SCCs. Each SCC consists of a two-node directed cycle, with a single root SCC and $m$ leaf SCCs. For each leaf SCC, we add exactly one directed edge from a randomly chosen node in the root SCC to a randomly chosen node in the leaf SCC. Interventions are restricted to leaf SCCs and are applied only to the node within each leaf SCC that does not receive the incoming edge from the root SCC. For a fixed number of interventions $|\mathcal{I}| = k$, we randomly select $k$ distinct leaf SCCs and apply one hard intervention per selected SCC. For each configuration, we evaluate whether the model parameters are identifiable up to a global scaling by analyzing the stacked linear system induced by the stationary first- and second-moment equations across all observational and interventional settings. Identifiability is assessed numerically via a spectral-gap criterion where we compute the singular values of the stacked system and declare identifiability if there is a clear separation between the smallest singular value and the remaining spectrum, corresponding to a one-dimensional nullspace. In all experiments, we used a threshold of $10^{10}$ for the ratio between the second-smallest and smallest singular values. Results are averaged over 30 random instances for each value of $k$.

Figure 2 shows the fraction of identifiable instances as a function of the number of interventions. We observe a sharp transition from non-identifiability to full identifiability when the number of interventions becomes equal to the number of leaf components (The dashed vertical lines shows the number of leaf SCCs). This result indicates that the theoretical bound from Theorem 3.6 is close to tight in this setting.

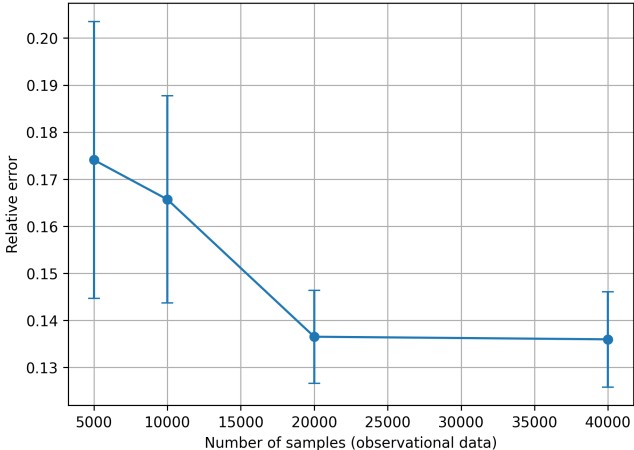

*Figure 3.* The average relative error for recovering $\mathbf{\Lambda}$ (up to some global scaling) versus the sample size (observational data).

### B.4. Effect of sample size on relative error

To assess the sensitivity of our estimator to the number of steady-state samples, we conducted an experiment in which we varied the number of observational samples used to estimate $(\hat{\boldsymbol{\mu}}, \hat{\boldsymbol{\Sigma}})$. For each trial, the number of samples per intervention was set to half of the observational sample size. We evaluated the relative error of recovering $\mathbf{\Lambda}$ for observational sample sizes $\{5\mathrm{k}, 10\mathrm{k}, 20\mathrm{k}, 40\mathrm{k}\}$ over 30 random instances with $n = 10$. As shown in Fig. 3, the estimation error decreases markedly when increasing the number of samples from 5k to 20k and then stabilizes, indicating that the estimator achieves accurate recovery with a moderate number of steady-state samples. This behavior is expected, since both empirical means and covariances become sufficiently accurate in this regime, after which additional samples yield diminishing returns.

### B.5. Finite-sample SCC recovery

We evaluate the SCC-recovery procedure implied by Theorem 3.3 and Remark 3.4 in a finite-sample setting. For each trial, we generate a random OU system with $n = 10$ nodes. The nodes are first partitioned into random SCCs of size between 1 and 3. The condensation graph is then generated as a random DAG by allowing edges only from earlier SCCs to later SCCs, with edge probability 0.3. Within each non-singleton SCC, we include a directed cycle to ensure strong connectivity and add extra within-SCC edges with probability 0.2. For each edge in the condensation DAG, we add one randomly selected inter-SCC edge.

The drift matrix $\mathbf{\Lambda}$ is made positive stable by strict row diagonal dominance: after assigning all off-diagonal weights, we set $\Lambda_{ii} = \sum_{j \neq i} |\Lambda_{ij}| + 0.5$. The input vector is sampled as $b_i \sim \mathrm{Unif}[0.5, 1.5]$, and the diagonal diffusion entries are sampled as $d_i \sim \mathrm{Unif}[0.2, 0.5]$. The observational stationary mean and covariance are computed from $\boldsymbol{\mu} = \mathbf{\Lambda}^{-1}\mathbf{b}$ and the Lyapunov equation $\mathbf{\Lambda}\mathbf{\Sigma} + \mathbf{\Sigma}\mathbf{\Lambda}^{\top} = \mathbf{D}$.

We use one intervention target per SCC, chosen as one representative node from that SCC. For hard interventions, we zero all off-diagonal entries in the intervened row while keeping the diagonal entry fixed. For soft interventions, we multiply the off-diagonal entries in the intervened row by a factor $\alpha = 0.4$, again keeping the diagonal entry fixed. For both observational and interventional settings, samples are drawn from the corresponding stationary Gaussian distributions.

To estimate the response set $R_i = \{j : \mu_j^{(i)} \neq \mu_j\}$, we perform coordinate-wise two-sample $z$-tests comparing observational and interventional samples. We then apply Benjamini–Hochberg correction with FDR level 0.01. The estimated response sets are used to reconstruct SCCs as follows: empty response sets are treated as singleton source SCCs; identical non-empty response sets are grouped together; strict inclusions among response sets recover the SCC-level partial order; and subtracting descendant response sets recovers the individual SCC node sets.

We repeat this procedure over 100 independently generated OU systems. The quality of the recovered SCC partition is measured by the Adjusted Rand Index (ARI), where 1 indicates exact recovery. We report results for both hard and soft interventions as a function of sample size.

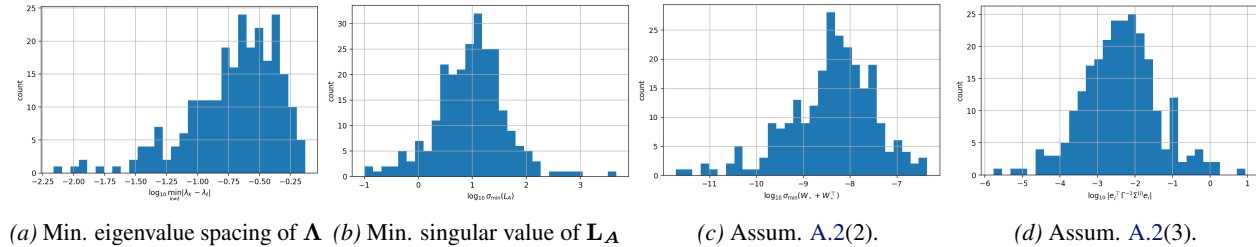

*(a)* Min. eigenvalue spacing of $\mathbf{\Lambda}$    *(b)* Min. singular value of $\mathbf{L_A}$      *(c)* Assum. A.2(2).      *(d)* Assum. A.2(3).

*Figure 4.* Numerical probe of spectral/rank assumptions: a) The histogram of minimum pairwise eigenvalue spacing of $\mathbf{\Lambda}$. b) The histogram of minimum singular of $\mathbf{L_A}$. c) The condition in Assumption A.2(2) d) The condition in Assumption A.2(3) (The values on the x-axis are in $\log_{10}$ scale)

## B.6. Experiment with non-diagonal diffusion

In this experiment, the data generator was modified to sample a full positive definite diffusion matrix $\mathbf{\Omega}$ instead of a diagonal $\mathbf{D}$. Specifically, we generated a random matrix $\mathbf{A}$, formed $\mathbf{AA}^\top$, normalized its scale, and added a positive diagonal jitter. The steady-state covariance was then obtained by solving the full Lyapunov equation $\mathbf{\Lambda\Sigma} + \mathbf{\Sigma\Lambda}^\top = \mathbf{\Omega}$. Thus, the data-generating process no longer satisfies the diagonal diffusion assumption. The learning model was also modified to estimate a full positive definite diffusion matrix. We parameterized $\mathbf{\Omega}$ as $\mathbf{\Omega} = \mathbf{MM}^\top + \epsilon\mathbf{I}$, where $\mathbf{M}$ is a learned lower-triangular matrix. Accordingly, the covariance loss was changed from separate diagonal and off-diagonal residuals to the full matrix residual $\mathbf{\Lambda}^{(c)}\widehat{\mathbf{\Sigma}}^{(c)} + \widehat{\mathbf{\Sigma}}^{(c)}\mathbf{\Lambda}^{(c)\top} - \mathbf{\Omega}$, summed over the observational and interventional contexts $c$. These changes ensure that both data generation and optimization use a full diffusion matrix, rather than treating diffusion as diagonal.

## C. About the Spectral/Rank Assumptions

### C.1. Numerical Analysis of Spectral/Rank Assumptions

We assessed the spectral nondegeneracy assumptions by Monte Carlo analysis on random, strongly connected OU models. In each of 250 trials, we sampled a drift matrix $\mathbf{\Lambda} \in \mathbb{R}^{10\times10}$ with off–diagonal pattern drawn i.i.d. at density 0.3 (Gaussian weights, standard deviation 0.8), and set each diagonal entry strictly larger than the $\ell_1$ row sum (ensuring Hurwitz stability). Strong connectivity was checked on the directed graph with edge $j \to i$ iff $\Lambda_{ij} \neq 0$. We drew a diagonal diffusion $\mathbf{D}$ with $d_k \sim \text{Unif}[0.2, 0.4]$, selected an intervention coordinate $i$ uniformly at random, and formed the intervened drift by zeroing row $i$'s off–diagonals (keeping $\Lambda_{ii}$). We then solved the continuous–time Lyapunov equations to obtain the observational and intervened covariances $\mathbf{\Sigma}$ and $\mathbf{\Sigma}^{(i)}$, formed $\mathbf{\Gamma} := \mathbf{\Sigma} - \mathbf{\Sigma}^{(i)}$, and built the $(-i, -i)$ Schur blocks $\mathbf{\Xi}$ and $\mathbf{Z}$ used in the proofs, with $\mathbf{A} := \mathbf{\Xi}^{-1}\mathbf{Z}$. We evaluated four quantities: (i) the simple-spectrum condition for $\mathbf{\Lambda}$, measured by $\min_{k\neq\ell}|\lambda_k - \lambda_\ell|$; (ii) the single-SCC skew-kernel condition in Assumption A.1, measured by $\sigma_{\min}(\mathbf{L_A})$, where $\mathbf{L_A}$ represents $\mathbf{S} \mapsto \text{offdiag}(\mathbf{SA} - \mathbf{A}^\top\mathbf{S})$ on the skew-symmetric subspace; (iii) the condition in Assumption A.2(2) and (iv) the condition in Assumption A.2(3), measured by $|\mathbf{e}_i^\top\mathbf{\Gamma}^{-1}\mathbf{\Sigma}^{(i)}\mathbf{e}_i|$. We report the resulting histograms in Figure 4. In all 250 trials, the four quantities were strictly separated from zero, supporting that the corresponding nondegeneracy assumptions hold.

### C.2. About the role of spectral/rank assumptions in the proofs

- Assumption A.1 holds on a nonempty open subset of the ambient $\mathbf{A}$-space. For example, if $\mathbf{A} = \text{diag}(a_1, \ldots, a_m)$ with $a_p \neq a_q$ for $p \neq q$, then for any skew-symmetric $\mathbf{S}$, $(\mathbf{SA} - \mathbf{A}^\top\mathbf{S})_{pq} = (a_q - a_p)S_{pq}$ for $p \neq q$. Hence offdiag$(\mathbf{SA} - \mathbf{A}^\top\mathbf{S}) = 0$ implies $S_{pq} = 0$ for all $p \neq q$, and since $\mathbf{S}$ is skew-symmetric, this gives $\mathbf{S} = 0$. Moreover, this injectivity is stable under sufficiently small off-diagonal perturbations. Indeed, write $\mathbf{A} = \mathbf{D} + \mathbf{E}$, where $\mathbf{D} = \text{diag}(a_1, \ldots, a_m)$ and $\delta := \min_{p\neq q}|a_p - a_q| > 0$. For any skew-symmetric $\mathbf{S}$, $\|\text{offdiag}(\mathbf{SD} - \mathbf{DS})\|_F \geq \delta\|\mathbf{S}\|_F$, whereas $\|\text{offdiag}(\mathbf{SE} - \mathbf{E}^\top\mathbf{S})\|_F \leq 2\|\mathbf{E}\|_2\|\mathbf{S}\|_F$. Therefore, if $2\|\mathbf{E}\|_2 < \delta$, then offdiag$(\mathbf{SA} - \mathbf{A}^\top\mathbf{S}) = 0$ forces $\mathbf{S} = 0$. Thus Assumption A.1 holds not only for diagonal matrices with separated diagonal entries, but also for all sufficiently small off-diagonal perturbations of such matrices. In the model, however, $\mathbf{A}$ is determined by the induced moments, so we state the condition directly as a checkable spectral condition on the induced operator $\mathcal{L}_\mathbf{A}$. The perturbation argument above suggests a direction for finding a witnesses. For instance, one may look for parameter regimes in a single SCC with suitably tuned weights, where the moment-derived matrix $\mathbf{\Xi}^{-1}\mathbf{Z}$ becomes close to a diagonal matrix with separated diagonal entries. In such a

regime, Assumption A.1 holds. For small SCCs the condition simplifies. When $|C| = 2$, we have $m = 1$, so the condition is vacuous. When $|C| = 3$, we have $m = 2$, and the condition reduces to the scalar requirement $A_{22} \neq A_{11}$.

- Assumption A.2 is used to guarantee that the covariance difference $\mathbf{\Gamma} := \mathbf{\Sigma} - \mathbf{\Sigma}^{(i)}$ is invertible. The factors in $\mathbf{W}_\star$ have a PBH-type interpretation. Since the SCC is strongly connected, the pair $(\mathbf{\Lambda}_\star, \mathbf{e}_i)$ is structurally controllable, so generically $\boldsymbol{\rho}_{\star,k}^\top \mathbf{e}_i \neq 0$ for all $k$. Similarly, if the intervention is non-null, then $\mathbf{w}_\star \neq 0$, and since $\mathbf{\Sigma}^{(i)} \succ 0$, the vector $\mathbf{\Sigma}^{(i)} \mathbf{w}_\star$ is nonzero. The factors $\mathbf{w}_\star^\top \mathbf{\Sigma}^{(i)} \boldsymbol{\rho}_{\star,\ell}$ can be viewed as PBH-type observability factors for the transpose dynamics. Strong connectivity of the SCC makes the corresponding structural observability condition generic. Thus the diagonal factors in $\mathbf{W}_\star$ are generically nonzero, while the assumption $0 \notin \sigma(\mathbf{W}_\star + \mathbf{W}_\star^\top)$ rules out the remaining symmetrized cancellation that makes $\mathbf{\Gamma}$ singular.

- The scalar condition $\mathbf{e}_i^\top \mathbf{\Gamma}^{-1} \mathbf{\Sigma}^{(i)} \mathbf{e}_i \neq 0$ is used in the proof to pass from invertibility of $\mathbf{\Gamma} := \mathbf{\Sigma} - \mathbf{\Sigma}^{(i)}$ to invertibility of the Schur-type matrix $\mathbf{\Xi}$. Let us see this condition on directed cycles. Suppose the SCC contains the cycle $1 \to 2 \to \cdots \to n \to 1$, and the intervention is on node 1. The hard intervention removes the edge $n \to 1$, so the interventional graph contains the directed chain $1 \to 2 \to \cdots \to n$. Thus the covariance effect generated by the intervention can propagate through all coordinates of the block. By Cramer's rule,

$$\mathbf{e}_1^\top \mathbf{\Gamma}^{-1} \mathbf{\Sigma}^{(1)} \mathbf{e}_1 = \frac{\det\left[\mathbf{\Sigma}^{(1)} \mathbf{e}_1, \mathbf{\Gamma} \mathbf{e}_2, \ldots, \mathbf{\Gamma} \mathbf{e}_n\right]}{\det(\mathbf{\Gamma})}.$$

On the directed cycles, both the numerator and denominator are rational functions of the cycle weights, self-loops, and diffusion variances. Since the intervened graph contains a directed chain from the intervened node to all other nodes, the columns appearing in the Cramer numerator are not forced to vanish by the graph structure. However, this does not by itself prove that the determinant is nonzero. To prove genericity on directed cycles, or on general SCCs, one would still need to exhibit one admissible parameter value for which the condition holds.

**Witness for directed cycle** For directed cycles, one can avoid the auxiliary $\mathbf{A} = \mathbf{\Xi}^{-1} \mathbf{Z}$ skew-kernel argument used in the general single-SCC proof. Instead, we work directly with the $\mathbf{\Xi}$-equation $\mathbf{\Lambda}_{\Delta,-i,-i} \mathbf{\Xi} + \mathbf{\Xi}^\top \mathbf{\Lambda}_{\Delta,-i,-i}^\top = 0$, which was derived in equation 28. When the SCC is the directed cycle $1 \to 2 \to \cdots \to n \to 1$ and the intervention is on node 1, the hard intervention removes the edge $n \to 1$. Hence, on the remaining nodes $2, \ldots, n$, the support of $\mathbf{X} := \mathbf{\Lambda}_{\Delta,-1,-1}$ is lower bidiagonal, corresponding to the directed chain $2 \to 3 \to \cdots \to n$. For this special support pattern, the adjacent-minor condition on $\mathbf{\Xi}$ given below forces $\mathbf{X} = 0$. Therefore, for directed cycles, the following argument provides a more concrete justification for the injectivity step that is handled in the general proof by the nondegeneracy assumption, Assumption A.1. This directed-cycle construction is also informative for SCCs that contain a directed cycle passing through all vertices in that SCC. Since our admissible parameter space allows the off-diagonal support of $\mathbf{\Lambda}$ to be a subset of $G$, one may restrict to this spanning-cycle subgraph by setting all remaining allowed edges to zero. Thus the spanning directed cycle gives an admissible subfamily on which the relevant $\mathbf{\Xi}$-based conditions can be examined explicitly.

Consider the directed cycle $1 \to 2 \to \cdots \to n \to 1$, with intervention on node 1. After the hard intervention, the edge $n \to 1$ is removed, and the remaining support on nodes $2, \ldots, n$ is the directed chain $2 \to 3 \to \cdots \to n$. Thus the ambiguity $\mathbf{X} := \mathbf{\Lambda}_{\Delta,-1,-1}$ has lower-bidiagonal support in the natural ordering $2, \ldots, n$. That is, if $m = n - 1$, then $X_{pq} = 0$ unless $p = q$ or $p = q + 1$.

Recall that the proof gives the equation $\mathbf{X} \mathbf{\Xi} + \mathbf{\Xi}^\top \mathbf{X}^\top = 0$. The following lemma gives a direct condition under which this equation forces $\mathbf{X} = 0$.

**Lemma C.1.** *Let $\mathbf{\Xi} \in \mathbb{R}^{m \times m}$ be arbitrary, not necessarily symmetric. Suppose $\mathbf{X} \in \mathbb{R}^{m \times m}$ is lower bidiagonal: $X_{pq} = 0$ unless $p = q$ or $p = q + 1$. If $\mathbf{X} \mathbf{\Xi} + \mathbf{\Xi}^\top \mathbf{X}^\top = 0$, and $\Xi_{11} \neq 0$, and, for every $p = 1, \ldots, m - 1$, $\Delta_p(\mathbf{\Xi}) := \Xi_{pp} \Xi_{p+1,p+1} - \Xi_{p,p+1} \Xi_{p+1,p} \neq 0$, then $\mathbf{X} = 0$.*

*Proof.* Write $x_p := X_{pp}, p = 1, \ldots, m$, and $\ell_p := X_{p+1,p}, p = 1, \ldots, m - 1$. These are the only possibly nonzero entries of $\mathbf{X}$. Let $\mathbf{E} := \mathbf{X} \mathbf{\Xi} + \mathbf{\Xi}^\top \mathbf{X}^\top$. By assumption, $\mathbf{E} = 0$. The $(1,1)$-entry gives $0 = E_{11} = 2x_1 \Xi_{11}$. Since $\Xi_{11} \neq 0$, we get $x_1 = 0$.

Now suppose, inductively, that for some $p \in \{1, \ldots, m - 1\}$,

$$x_1 = \cdots = x_p = 0, \qquad \ell_1 = \cdots = \ell_{p-1} = 0.$$

Then row $p$ of $\mathbf{X}$ is zero, while row $p+1$ has only two possibly nonzero entries, $\ell_p$ in column $p$ and $x_{p+1}$ in column $p+1$. Hence the $(p, p+1)$-entry of $\mathbf{E} = 0$ gives

$$\ell_p \Xi_{pp} + x_{p+1} \Xi_{p+1,p} = 0,$$

and the $(p+1, p+1)$-entry gives

$$\ell_p \Xi_{p,p+1} + x_{p+1} \Xi_{p+1,p+1} = 0.$$

Therefore,

$$\begin{bmatrix} \Xi_{pp} & \Xi_{p+1,p} \\ \Xi_{p,p+1} & \Xi_{p+1,p+1} \end{bmatrix} \begin{bmatrix} \ell_p \\ x_{p+1} \end{bmatrix} = 0.$$

Its determinant is

$$\Delta_p(\mathbf{\Xi}) = \Xi_{pp} \Xi_{p+1,p+1} - \Xi_{p,p+1} \Xi_{p+1,p},$$

which is nonzero by assumption. Thus $\ell_p = 0$ and $x_{p+1} = 0$. The claim follows by induction. $\qquad \square$

For a directed cycle, define

$$\kappa_{\Xi} := \Xi_{11} \prod_{p=1}^{m-1} \left( \Xi_{pp} \Xi_{p+1,p+1} - \Xi_{p,p+1} \Xi_{p+1,p} \right).$$

By Lemma C.1, if $\kappa_{\Xi} \neq 0$, then the $\mathbf{\Xi}$-equation forces the directed-chain ambiguity $\mathbf{\Lambda}_{\Delta,-1,-1}$ to vanish.

We now show that $\kappa_{\Xi}$ is not identically zero on the directed-cycle parameter space, for every cycle length $n$. Since $\mathbf{\Xi}$ is obtained from solutions of Lyapunov equations, its entries are rational functions of the cycle weights, self-loops, and diffusion variances. Hence each factor in $\kappa_{\Xi}$ is a rational function of the model parameters.

We use induction on $n$. The base cases are direct. For $n = 2$, take

$$\mathbf{\Lambda} = \begin{bmatrix} 2 & 1 \\ 1 & 2 \end{bmatrix}, \qquad \mathbf{D} = \mathrm{diag}(1, 2).$$

A direct calculation gives

$$\mathbf{\Xi} = \begin{bmatrix} \dfrac{3}{64} \end{bmatrix},$$

so $\kappa_{\Xi} \neq 0$. For $n = 3$, take

$$\mathbf{\Lambda} = \begin{bmatrix} 2 & 0 & 1 \\ 1 & 2 & 0 \\ 0 & 1 & 2 \end{bmatrix}, \qquad \mathbf{D} = \mathrm{diag}(1, 2, 3).$$

For the intervention on node 1, direct calculation gives

$$\mathbf{\Xi} = \begin{bmatrix} \dfrac{1}{4032} & -\dfrac{1}{8064} \\ \dfrac{19}{384} & -\dfrac{265}{21504} \end{bmatrix}.$$

Thus

$$\Xi_{11} = \frac{1}{4032} \neq 0, \qquad \det(\mathbf{\Xi}) = \frac{89}{28901376} \neq 0.$$

Hence $\kappa_{\Xi} \neq 0$ for the 3-cycle.

Assume now that the claim holds for the $n$-cycle. We prove it for the $(n+1)$-cycle. Let

$$P_p^{(n+1)}(\theta) := \Xi_{pp}^{(n+1)} \Xi_{p+1,p+1}^{(n+1)} - \Xi_{p,p+1}^{(n+1)} \Xi_{p+1,p}^{(n+1)}, \qquad p = 1, \ldots, n-1,$$

and let $P_0^{(n+1)}(\theta) := \Xi_{11}^{(n+1)}$. We will show that each $P_p^{(n+1)}$ is not identically zero as a rational function of the $(n+1)$-cycle parameters.

First consider $p = 0, \ldots, n-2$, i.e., the factors involving only the old reduced nodes $2, \ldots, n$. By the induction hypothesis, choose an admissible $n$-cycle whose corresponding $\mathbf{\Xi}^{(n)}$-factors are all nonzero. We now insert node $n+1$ as a fast relay between $n$ and 1.

Let $\mathbf{\Lambda}^{\mathrm{old}} \in \mathbb{R}^{n \times n}$ be the drift of the old $n$-cycle, and let $\omega := \Lambda_{1n}^{\mathrm{old}}$ be the closing-edge weight $n \to 1$. Define $\mathbf{A} := \mathbf{\Lambda}^{\mathrm{old}} - \omega \mathbf{e}_1 \mathbf{e}_n^\top$, so that $\mathbf{A}$ is the old drift with the closing edge removed. Write $\mathbf{x} := (x_1, \dots, x_n)^\top$, $z := x_{n+1}$. For parameters $\alpha, \beta, c > 0$ satisfying $\frac{\alpha\beta}{c} = \omega$, define the relay drift

$$\mathbf{\Lambda}_\varepsilon = \begin{bmatrix} \mathbf{A} & \beta\mathbf{e}_1 \\ -\alpha\varepsilon^{-1}\mathbf{e}_n^\top & c\varepsilon^{-1} \end{bmatrix}.$$

The relay coordinate has drift $-\frac{c}{\varepsilon}z + \frac{\alpha}{\varepsilon}x_n = -\frac{c}{\varepsilon}\left(z - \frac{\alpha}{c}x_n\right)$, so as $\varepsilon \to 0$, the relay enforces $z \approx \frac{\alpha}{c}x_n$. Therefore the edge $n+1 \to 1$ contributes $-\beta z \approx -\frac{\alpha\beta}{c}x_n = -\omega x_n$, which is exactly the contribution of the old closing edge $n \to 1$.

We verify this at the covariance level. Let

$$\mathbf{\Sigma}_\varepsilon = \begin{bmatrix} \mathbf{S}_\varepsilon & \mathbf{r}_\varepsilon \\ \mathbf{r}_\varepsilon^\top & s_\varepsilon \end{bmatrix}$$

be the stationary covariance of $(\mathbf{x}, z)$, where $\mathbf{S}_\varepsilon = \mathrm{Cov}(\mathbf{x}, \mathbf{x})$, $\mathbf{r}_\varepsilon = \mathrm{Cov}(\mathbf{x}, z)$. Assume the diffusion matrix has block form

$$\mathbf{D}_\varepsilon = \begin{bmatrix} \mathbf{D}_x & 0 \\ 0 & d_z \end{bmatrix}.$$

The $(\mathbf{x}, z)$-block of the Lyapunov equation gives

$$\mathbf{A}\mathbf{r}_\varepsilon + \beta\mathbf{e}_1 s_\varepsilon - \alpha\varepsilon^{-1}\mathbf{S}_\varepsilon\mathbf{e}_n + c\varepsilon^{-1}\mathbf{r}_\varepsilon = 0.$$

Multiplying by $\varepsilon$ and rearranging,

$$\mathbf{r}_\varepsilon = \frac{\alpha}{c}\mathbf{S}_\varepsilon\mathbf{e}_n - \frac{\varepsilon}{c}\left(\mathbf{A}\mathbf{r}_\varepsilon + \beta\mathbf{e}_1 s_\varepsilon\right).$$

The covariance blocks remain bounded for sufficiently small $\varepsilon$, since the relay has a fast stable self-loop $c/\varepsilon$ and the effective slow drift is $\mathbf{\Lambda}^{\mathrm{old}}$, which is positive stable. Hence

$$\mathbf{r}_\varepsilon = \frac{\alpha}{c}\mathbf{S}_\varepsilon\mathbf{e}_n + O(\varepsilon).$$

The $(\mathbf{x}, \mathbf{x})$-block of the Lyapunov equation is

$$\mathbf{A}\mathbf{S}_\varepsilon + \mathbf{S}_\varepsilon\mathbf{A}^\top + \beta\mathbf{e}_1\mathbf{r}_\varepsilon^\top + \beta\mathbf{r}_\varepsilon\mathbf{e}_1^\top = \mathbf{D}_x.$$

Substituting the previous relation gives

$$\left(\mathbf{A} + \frac{\alpha\beta}{c}\mathbf{e}_1\mathbf{e}_n^\top\right)\mathbf{S}_\varepsilon + \mathbf{S}_\varepsilon\left(\mathbf{A} + \frac{\alpha\beta}{c}\mathbf{e}_1\mathbf{e}_n^\top\right)^\top = \mathbf{D}_x + O(\varepsilon).$$

Since $\alpha\beta/c = \omega$, the drift in parentheses is

$$\mathbf{A} + \omega\mathbf{e}_1\mathbf{e}_n^\top = \mathbf{\Lambda}^{\mathrm{old}}.$$

Therefore, by uniqueness and continuity of Lyapunov solutions,

$$\mathbf{S}_\varepsilon \longrightarrow \mathbf{\Sigma}^{\mathrm{old}} \qquad \text{as } \varepsilon \to 0.$$

The same argument applies to the intervened covariance. Under the intervention on node 1, the edge $n+1 \to 1$ is removed, so the block $\beta\mathbf{e}_1$ is set to zero. Thus the effective closing edge also disappears, matching the intervened old $n$-cycle. Hence the observational and interventional covariance blocks on the old variables converge to those of the old $n$-cycle.

Consequently, the corresponding old block of $\mathbf{\Xi}_\varepsilon$ converges to the old $\mathbf{\Xi}^{(n)}$-matrix. Therefore, for $p = 0, \dots, n-2$, the factors $P_p^{(n+1)}$ converge to the nonzero factors of the old $n$-cycle. Hence each $P_p^{(n+1)}$, $p = 0, \dots, n-2$, is not identically zero as a rational function of the $(n+1)$-cycle parameters.

It remains to show that the last factor

$$P_{n-1}^{(n+1)} = \Xi_{n-1,n-1}^{(n+1)}\Xi_{n,n}^{(n+1)} - \Xi_{n-1,n}^{(n+1)}\Xi_{n,n-1}^{(n+1)}$$

is not identically zero. For this factor, use a different relay limit. Collapse the path

$$1 \to 2 \to \cdots \to n$$

to a single effective edge $1 \to n$ by making the intermediate nodes $2, \ldots, n-1$ fast relays. The $(n+1)$-cycle then converges, on the retained nodes $\{1, n, n+1\}$, to a directed 3-cycle

$$1 \to n \to n+1 \to 1.$$

Under this collapse, the last adjacent $\Xi$-minor of the $(n+1)$-cycle converges to the unique adjacent $\Xi$-minor of this effective 3-cycle. By the explicit 3-cycle calculation above, this minor is nonzero for an admissible choice of the effective 3-cycle parameters. Therefore $P_{n-1}^{(n+1)}$ is not identically zero.

We have shown that every factor $P_p^{(n+1)}$, $p = 0, \ldots, n-1$, is a nonzero rational function. Since the product of finitely many nonzero rational functions is again nonzero, we conclude that

$$P_0^{(n+1)} P_1^{(n+1)} \cdots P_{n-1}^{(n+1)}$$

is not identically zero. Hence there exists an admissible $(n+1)$-cycle parameter value for which all adjacent $\Xi$-minors are nonzero. This completes the induction.

Consequently, for every directed cycle length $n$, the adjacent-$\Xi$ condition holds generically on the directed-cycle parameter subfamily. By Lemma C.1, the equation

$$\mathbf{\Lambda}_{\Delta,-1,-1}\mathbf{\Xi} + \mathbf{\Xi}^\top \mathbf{\Lambda}_{\Delta,-1,-1}^\top = 0$$

has no nonzero solution with directed-chain support, generically on this subfamily.

