# OpenReview forum: "One Intervention per Component is Enough: Towards Identifiability in Linear Stochastic Dynamics from Steady State"
_ICML.cc/2026/Conference — ICML 2026 spotlight_

### Official Review · Reviewer_j3JN · 2026-03-01

**Soundness:** 3
**Presentation:** 3
**Significance:** 3
**Originality:** 3
**Overall Recommendation:** 5
**Confidence:** 5

**Summary:**

The paper studies parameter identifiability for multivariate Ornstein-Uhlenbeck (OU) processes, using only "snapshot" data (i.e., the stationary distribution), from both observational and interventional distributions. In particular, they consider an OU process with a positive stable drift matrix $\Lambda$ (so that the process has a stationary distribution) and diagonal diffusion matrix $\sigma$. For interventions, they consider a special form of single-node "hard"/"perfect" interventions which preserves self-regulation (i.e., an intervention on $i$ sets $\Lambda_{ik} = 0$ for $k \neq i$ and leaves all other drift/diffusion parameters unchanged).

They prove that, up to a global scaling factor, the parameters are identifiability under certain graphical, interventional, and non-degenericity conditions, including (1) a connected graph with a single strongly connected component (SCC) as the "root", (2) an intervention in each SCC, and (3) conditions on the spectra of some moment matrices. Then, they propose a learning algorithm based on a simple least-squares objective, which they test in experiments on synthetic and real-world data.

In the synthetic experiments, they show that their method outperforms a related baseline (which minimizes an objective equivalent to their own with certain terms removed) and that relative error decreases as the number of interventions increases. In the real-world experiments, they apply their method to Perturb-seq data with CRISPR knockouts and evaluate their method on generalization to unseen perturbations. They show that their method outperforms the observational baseline, and achieves comparable and sometimes better performance to an oracle baseline.

**Compliance With Llm Reviewing Policy:**

Affirmed.

**Final Justification:**

The authors' responses indicate plans for revisions that will improve the paper, and I appreciate their efforts. Overall, their responses have fully addressed my concerns, and I am comfortable recommending acceptance.

The limiting assumption of an OU process is the main weakness that could limit the impact of the paper from the methodological perspective, but I agree with the authors that "it is important to first establish identifiability results in linear SDEs, where the analysis is more tractable and the guarantees are more interpretable". In my experience, such identifiability results can likely be generalized to nonparametric settings (as the authors already mention for SCC recovery). Hence, the paper could have more substantial impact from the theoretical perspective (in agreement with `GcTM`), and this potential impact is enough for me to raise my score from a 4 to a (low) 5.

**Key Questions For Authors:**

1. **Discussion of assumptions:** As referred to in Weakness #1 (and somewhat in Minor Weakness #4), the assumptions are somewhat restrictive, and there are several directions in which they could be generalized (e.g. to a more nonparametric setting, to multi-node interventions, to soft interventions). Can you discuss how you envision your insights/proof techniques being generalized to these settings? My view of the paper would be improved if some of the insights are transferable.
2. **Hyperparameter selection:** As referred to in Weakness #3, please provide the exact hyperparameter values used for the experiments, and a description of how they are determined. A fair determination of the hyperparameters is critical for the soundness of the experimental results - my evaluation of the paper hinges on the soundness of these results, and I will raise my score if the procedure is reasonable (or better yet, if the results are consistent across a range of hyperparameters).

**Limitations:**

The authors could better address the restrictiveness of their assumptions; otherwise, the limitations are adequately discussed.

**Strengths And Weaknesses:**

## Strengths
1. **Motivation:** The paper is motivated by a real scientific problem with broad applicability, and the experiments tie back to such an application.
2. **Real-world evaluation:** The real-world experiments use appropriate, task-specific metrics that are common for perturbation prediction.
3. **Clear setup and motivated assumptions:** The paper clearly lays out their mathematical assumptions in Section 2 and clearly defines terms such as "strongly connected components", "condensation graph", etc. The graphical and interventional assumptions are somewhat restrictive/stylized (see Weakness 1 below), but otherwise mostly well-motivated, e.g. the authors note that the form of the interventions is "analogous to a knockout perturbation which isolates a variable from its regulators", and they provide experiments in the appendix to show that their non-degeneracy assumptions hold frequently in sampled instances.
4. **Presentation and related work:** The paper is well-structured, proceeding in a logical order through problem formulation, identifiability results, a learning algorithm, and experiments. The related work section is well-written and informative, adequately positioning itself with respect to existing literature on identifiability of OU processes (or SDEs more generally), and providing an important distinction between identifiability-driven results/algorithms vs. predictive steady-state perturbation modeling.

## Weaknesses
1. **Restrictive/stylized assumptions:** In general, OU processes are quite restrictive compared to general diffusion processes, requiring an affine, time-invariant drift function and a linear, time-invariant diffusion matrix. However, I do appreciate its central importance in many scientific applications - it is important in its own right, and a reasonable starting point for follow-on work. On the graphical side, the "single root SCC" assumption is a bit limiting, and the authors do not provide much explanation for its relevance in applications (i.e., it seems primarily to be assumed so that the proof goes through). On the intervention side, the form of the intervention is better motivated, but quite restrictive compared to more general "soft" interventions which could arbitrarily change the drift coefficients/diffusion term of the intervened node. The abstract points out the graphical and non-degeneracy assumptions, but should also point out the intervention assumptions (single-node, a specific form of self-regulation preserving "hard" intervention).
2. **Synthetic data details:** The authors provide many of the experiment details in Appendix B, some of which are important enough to be described in the main text, e.g. the number of samples per distribution. Furthermore, some details do not seem to be addressed at all, e.g. are the intervention targets chosen at random?
3. **Missing hyperparameter specification:** The learning objective in Equations (10) and (11) has three hyperparameters ($\alpha_O$, $\alpha_I$, $\gamma$), but the authors do not specify what values are used or how these are selected, e.g. are they cherry-picked, or are they selected by cross-validation? This detail is especially important to interpret the results in Figure 1, where the comparison between "mean+covariance" and "covariance-only" depends directly on which terms are emphasized/deemphasized.

### Minor Weaknesses
1. **Description of OU process:** In the problem formulation, the paper should point to a definition of "positive stable" and discuss why it is assumed (to guarantee a stationary distribution). Then, a reference or proof should be given for the mean and covariance of the stationary distribution. As a minor note, $W_t$ is the Weiner process, not $dW_t$.
2. **Algorithm 1:** To my understanding, it seems that Algorithm 1 is intended as a proof device, rather than as a practical algorithm (which would instead be to minimize the least-squares objective). It may be helpful to further emphasize this, if correct.
3. **Figure 1 caption:** It would be helpful for the Figure 1 caption to be more detailed, to make the figure more self-contained, e.g. distinguishing "Mean + Covariance" from "Covariance" and specifying $n=10$.
4. **Related work on interventional causal discovery:** The causal structure learning / causal discovery literature has very well-developed theory for identifiability from interventions, e.g. for (possibly multi-node) soft interventions in nonparametric settings [1], with unknown targets [2], and these results also provide "partial identifiability" when there are not sufficient interventions for complete identifiability. I appreciate that the related work focuses on SDE models, which avoids overloading the reader with context, but it may be helpful in the discussion to point out more directions for future work and use the causal discovery literature as an analogy/motivation for more general results.


[1] Yang et al. (2018), "Characterizing and learning equivalence classes of causal dags under interventions"
\
[2] Squires et al. (2020), "Permutation-based causal structure learning with unknown intervention targets"

---

> ### Author Rebuttal · Authors · 2026-03-28
>
> > Restrictive/stylized assumptions
>
> On the graphical assumptions, we would like to clarify their roles more explicitly. For recovering the SCCs and the DAG over SCCs (Theorem 3.3), the main requirement is that there is at least one intervention per SCC (the main message of the paper). This assumption is plausible in settings where interventions can target different parts of the system, ensuring that each SCC is sufficiently perturbed. For identifiability of the full parameters of the SDE (up to a global scaling), additional assumptions are required. In particular, we assume that there is a single root SCC; otherwise, as mentioned in Remark 3.6, it is not possible to identify all parameters under a common global scaling when multiple root components exist. In addition, we require certain non-degeneracy spectral conditions, which serve to rule out degenerate algebraic cases and are expected to hold generically.
>
> Our choice of “hard” interventions (zeroing off-diagonal entries in a row while preserving self-regulation) is motivated by standard formulations in causal SDEs and enables clean identifiability results. As stated in the paper (page 2, right column, lines 75–80), this is analogous to a knockout perturbation that isolates a variable from its regulators, and similar formulations have been considered in prior work (e.g., Hansen & Sokol, 2014; Boeken & Mooij, 2024). However, our results on recovering SCCs and the DAG over SCCs can be extended to soft interventions, where the intervened row is perturbed rather than exactly zeroed. We will add the proof in the paper. In the rebuttal phase, we also performed experiments for recovering SCCs from soft interventions. Please see the experimental results given in the response to the reviewer GcTM.
>
> Finally, we agree that the intervention assumptions should be stated more explicitly in the abstract. In the revision, we will clarify the type of interventions considered (single-node, self-regulation-preserving hard interventions) and indicate which results extend to more general (soft) intervention settings.
>
> > Synthetic data details
>
> In the revised version (with additional page allowance), we will move key details into the main text to improve clarity. Regarding intervention selection, the intervention targets are chosen randomly. We will explicitly state this in the revised version to ensure all experimental details are clearly specified.
>
> > Missing hyperparameter specification
>
> We performed a grid search over the hyperparameters in the learning objective (Eqs. (10)–(11)), and for each approach (e.g., “mean+covariance” and “covariance-only”), we reported the results corresponding to the best-performing combination. In particular, we searched over $\alpha_I, \alpha_O, \gamma \in \\{0.01,0.1,1,10\\}$. We will include the full description of the hyperparameter search space and selection procedure in the revised version.
>
> > Description of OU process
>
> We will add a formal definition of positive stability in the revised version. In our formulation, $d\mathbf{x} = (-\boldsymbol{\Lambda}\mathbf{x} + \mathbf{b})\,dt + \boldsymbol{\sigma}\, d\mathbf{W}_t$,
> this means that all eigenvalues of $\boldsymbol{\Lambda}$ have strictly positive real parts (equivalently, $-\boldsymbol{\Lambda}$ is Hurwitz stable). This assumption guarantees that the OU process admits a unique stationary distribution. We will also clarify its role by noting that, given $\boldsymbol{D}$ being positive definite, the Lyapunov equation has a unique positive definite solution $\boldsymbol{\Sigma}$ if and only if $\boldsymbol{\Lambda}$ is positive stable as stated for example in Theorem 1.1 of (Frommer and Hashemi 2012).
>
> Regarding the mean and covariance of the stationary distribution, we will add a standard reference on Ornstein–Uhlenbeck processes, such as Øksendal, Stochastic Differential Equations.
>
> Finally, thanks for the minor note. $\mathbf{W}_t$ is indeed the Wiener process, and we will correct the wording in the paper.
>
> > Algorithm 1
>
> Yes, Algorithm 1 is primarily intended as a proof device to establish the identifiability results. We will emphasize this more in Remark 3.7.
>
> > Figure 1 caption
>
> In the revised version, we will expand the caption of Figure 1 to clearly distinguish between “Mean + Covariance” and “Covariance only,” and to specify all relevant experimental details so that the figure is more self-contained.
>
> > Related work on interventional causal discovery
>
> Due to space limitations, we focused the related work in the main text on identifiability results for SDEs to keep the presentation focused. We agree that connections to the broader causal discovery literature (e.g., intervention-based identifiability in SCMs) provide useful context. In the revised version, we will add a discussion in the appendix reviewing these works.

---

> > ### Author Rebuttal · Reviewer_j3JN · 2026-04-03
> >
> > Thank you for the thoughtful response! My points are mostly resolved, see below:
> >
> > **Restrictive assumptions:** Thank you for the clarification. I agree it would be helpful in the revisions to indicate which assumptions are necessary for different recovery guarantees, and I appreciate that the authors plan to state the intervention assumptions more explicitly in the appendix. From the response, my understanding is as follows:
> > - For recovering the strongly connected components (SCCs) and the DAG over the SCCs, one intervention per SCC is enough, without assuming a single root SCC or hard interventions. I completely agree that this "intervention diversity" condition is reasonable for many settings.
> > - The single root SCC assumption and the hard intervention assumption are required for identifiability of the parameters up to a single global scaling.
> > This division - especially with weak vs. hard interventions - resembles prior work in causal representation learning, so seems quite reasonable. I also appreciate the addition of experiments with soft interventions. Hence, I would consider this point fully resolved.
> >
> > **Synthetic data details:** This point is adequately resolved - I agree with the authors' plan to move key details to the main text in the revision.
> >
> > **Missing hyperparameter specification:** Thank you for clarifying the hyperparameter grid over which the hyperparameters were selected. However, the use of the "best-performing combination" seems to suggest that the hyperparameters were selected based on the test set, rather than a validation set, which is not best practice. Hence, this point is only partially resolved.
> >
> > **Description of OU process:** Thank you for agreeing to add a definition of positive stability and a citation for the mean/covariance. These were small details and adequately addressed.
> >
> > **Related work on interventional causal discovery:** I agree with the choice to keep the presentation focused on identifiability of SDEs, so the proposal (reviewing causal discovery in the appendix) strikes a good balance. I expect that researchers from causal discovery would be a prime audience for this paper, so such an addition will help orient that audience and attract more readers. Indeed, for this audience, it may be worth adding ~1 short paragraph in the main text. This point is adequately resolved.
> >
> > **Other points:** Thanks for confirming that Algorithm 1 is primarily a proof device and for the proposed expansion to the caption of Figure 1; these points are adequately addressed.
> >
> > Based on these responses, I maintain my "weak accept" recommendation - the changes will improve the paper, and it is certainly a solid technical contribution, but the fundamental assumption of an OU process (rather than a more nonparametric process) remains an important limitation on the overall impact.

---

> > > ### Author Response · Authors · 2026-04-03
> > >
> > > Thank you for the positive reassessment. We are glad that our clarifications helped resolve most of your concerns.
> > >
> > > Regarding hyperparameter tuning, to clarify, a portion of the training data was held out for validation, and hyperparameters were selected based on validation performance rather than on the test set. We omitted this detail in the rebuttal, as this is standard experimental practice, but we will state it explicitly in the revised version.
> > >
> > > Regarding the OU assumption, we believe it is important to first establish identifiability results in linear SDEs, where the analysis is more tractable and the guarantees are more interpretable, while still being meaningful in applications where linear approximations are often useful. In the revised version, we will make this motivation clearer and also mention extensions, in particular SCC recovery, to more general nonlinear settings.

---

### Official Review · Reviewer_sSYd · 2026-03-10

**Soundness:** 4
**Presentation:** 3
**Significance:** 3
**Originality:** 3
**Overall Recommendation:** 5
**Confidence:** 1

**Summary:**

The paper studies the identifiability of the parameters of a multivariate Ornstein–Uhlenbeck process when the full trajectories are not observed, and only snapshot observations are available. In addition to the steady-state parameters, the authors study how interventions affect these parameters and how such changes can be inferred. They provide assumptions under which the parameters are identifiable in both single- and multiple-intervention settings, and propose algorithms to perform the corresponding inference.

**Compliance With Llm Reviewing Policy:**

Affirmed.

**Final Justification:**

I put 5 because, reading through the paper, it seems a serious piece of work and an interesting topic. I am way out of my expertise and my judgment should matter little.

**Key Questions For Authors:**

It would be great to have a bit more understanding of the main assumptions in the main text.

**Limitations:**

Yes

**Strengths And Weaknesses:**

Strengths:
- The paper looks precise and technically sound.
- It is well written

Weaknesses
- This is way too much outside my expertise to genuinely find something meaningful.

---

> ### Author Rebuttal · Authors · 2026-03-28
>
> > It would be great to have a bit more understanding of the main assumptions in the main text.
>
> For recovering the SCCs and the DAG over SCCs (Theorem 3.3), the main requirement is that there is at least one intervention per SCC, ensuring that each component is sufficiently perturbed to reveal its structure. For identifiability of the full parameters (up to a global scaling), we additionally require a single root SCC, since otherwise the parameters cannot be identified under a common scaling across multiple root components, as discussed in Remark 3.6. Finally, the spectral assumptions serve to exclude degenerate algebraic cases. We will incorporate this explanation into the main text to make the assumptions more interpretable and easier to follow.

---

> > ### Author Rebuttal · Reviewer_sSYd · 2026-04-02
> >
> > Thanks!

---

### Official Review · Reviewer_zR47 · 2026-03-11

**Soundness:** 2
**Presentation:** 2
**Significance:** 1
**Originality:** 2
**Overall Recommendation:** 4
**Confidence:** 2

**Summary:**

This paper studies when a multivariate Ornstein–Uhlenbeck system can be identified from steady-state observational and intervention data, and shows that, generically, one intervention per strongly connected component is enough to recover the drift, input, and diffusion parameters up to a global scaling factor.

**Compliance With Llm Reviewing Policy:**

Affirmed.

**Final Justification:**

I still think the real data motivation is not enough, but the paper has some valuable contribution.

**Key Questions For Authors:**

Can the authors discuss the assumptions behind the main theorems more clearly, including their plausibility and examples of settings where they hold?

Can the authors provide more comprehensive simulations, at least in the appendix, covering a wider range of parameter settings?

Can the authors strengthen the real-data motivation and, if possible, include additional empirical analysis?

**Limitations:**

yes

**Strengths And Weaknesses:**

Strengths:

1-The paper appears theoretically sound.

Weaknesses:

1-The theoretical conditions appear rather restrictive. In particular, the assumptions of a single root strongly connected component and certain spectral non-degeneracy conditions may substantially limit the applicability of the results.

2-In addition, the real-data motivation and empirical example are not especially compelling. The application does not fully convince the reader of the practical importance or broad usefulness of the proposed method.

---

> ### Author Rebuttal · Authors · 2026-03-28
>
> > The theoretical conditions appear rather restrictive...
>
> Regarding the assumption of a single root SCC, we emphasize that this condition is only required for identifiability of the full parameters of the SDE (up to a global scaling). In fact, as discussed in Remark 3.6, we explicitly show that this assumption is necessary by providing an example with multiple root SCCs, where the parameters of the OU process are not identifiable up to a single global scaling. Moreover, this assumption is not required for our results on recovering the SCCs and the condensation graph over them (Theorem 3.3 and Remark 3.4).
>
> Regarding the spectral assumptions, we refer to our discussion in Appendix C.2. These assumptions mainly serve to rule out degenerate algebraic cases and are expected to hold generically.
>
> > In addition, the real-data motivation...
>
> The goal of our empirical evaluation is to demonstrate the practical implications of our identifiability results, in particular for predicting unseen perturbations, which is a central task in systems biology.
> Specifically, our real-data experiment focuses on gene perturbation prediction, where the objective is to infer the steady-state response of a system under interventions that were not observed during training. This is a challenging and practically important problem in computational biology, as exhaustive experimental perturbation is often infeasible. Our method leverages interventional steady-state data to recover the underlying structure and enables prediction under unseen interventions, which we demonstrate empirically.
>
> > Q1: Can the authors discuss the assumptions...
>
> For recovering the SCCs and the DAG over SCCs (Theorem 3.3), the main requirement is that there is at least one intervention per SCC (the main message of the paper). This assumption is plausible in settings where interventions can target different parts of the system, ensuring that each SCC is sufficiently perturbed. For identifiability of the full parameters of the SDE (up to a global scaling), additional assumptions are required. In particular, we assume that there is a single root SCC; otherwise, as discussed above, it is not possible to identify all parameters under a common global scaling when multiple root components exist. In addition, we require certain non-degeneracy spectral conditions, which serve to rule out degenerate algebraic cases.
>
> > Q2: Can the authors provide more...
>
> We thank the reviewer for this suggestion. However, this comment is quite broad, since “a wider range of parameter settings” is not specified and could refer to many different aspects of the model. In the current submission, we already studied several of the main dimensions that are most relevant to our setting: the number of interventions, graph density, and observational sample size. In particular, Figure 1 reports recovery performance as a function of the number of interventions and also examines how often the graphical conditions in Theorem 3.5 are satisfied under different graph densities. Appendix B.3 further studies the conservativeness of the graphical conditions, and Appendix B.4 analyzes the effect of observational sample size on recovery performance.
>
> > Q3: Can the authors strengthen the real-data motivation...
>
>  We will clarify more the motivation behind the real-data experiment, in particular by emphasizing its connection to unseen perturbation prediction (as we discussed above). Regarding the request for “additional empirical analysis,” the comment is somewhat broad, as it is not clear which specific aspects are missing. Nevertheless, to address the comments of the reviewer GcTM, we evaluated finite-sample recovery of the SCC decomposition under both hard and soft interventions. The results show that SCC recovery improves steadily with sample size. Second, we repeated the parameter recovery experiment in the same setup as Figure 1 but without imposing the diagonal diffusion assumption. The results show the same trend as in Figure 1, where the relative error decreases consistently as the number of interventions increases.
>
> We thank the reviewer for their evaluation. However, we respectfully note that the comments provided do not indicate issues that would correspond to a score of 2 (reject), such as technical flaws, lack of reproducibility, or unclear presentation. In particular, our work provides the first identifiability results for linear SDEs from steady-state data when the diffusion matrix is unknown, under explicit and well-motivated assumptions. These theoretical guarantees are complemented by empirical results. We have also ensured reproducibility by providing detailed experimental descriptions (Appendix B) and releasing code.  Given this, we would appreciate clarification on the aspects that justify the assigned score, as we do not see evidence of the issues typically associated with this rating.

---

> > ### Author Rebuttal · Reviewer_zR47 · 2026-04-01
> >
> > The authors have not fully addressed my concerns regarding the empirical evaluation. In particular, I do not believe that the real-data applications and simulation studies provide sufficient justification for the proposed method. That said, I appreciate the authors’ efforts to clarify several important aspects of the theoretical conditions, and in light of those clarifications, I will raise my score.

---

> > > ### Author Response · Authors · 2026-04-01
> > >
> > > We thank the reviewer for acknowledging the clarifications on the theoretical aspects. Regarding the empirical evaluation, we would like to emphasize that we already conducted several simulation studies in the original submission (varying the number of interventions, graph density, and sample size), and further added additional experiments in the rebuttal (e.g., finite-sample SCC recovery and robustness to non-diagonal diffusion) in response to other reviewers’ comments. Given this, we would appreciate more specific suggestions on what additional simulations would further strengthen the empirical section.

---

### Official Review · Reviewer_GcTM · 2026-03-12

**Soundness:** 3
**Presentation:** 4
**Significance:** 3
**Originality:** 3
**Overall Recommendation:** 5
**Confidence:** 3

**Summary:**

This paper studies identification of linear stochastic dynamics from steady-state data. The setting is a multivariate Ornstein–Uhlenbeck process with diagonal diffusion, where only stationary observational and interventional snapshots are available. The main theoretical claim is that, under a connected SCC condensation graph with a unique root and certain spectral nondegeneracy assumptions, one intervention per strongly connected component is generically sufficient to recover the drift, bias, and diagonal diffusion parameters up to a global scaling factor. The paper first proves a single-SCC identifiability result, then extends it to the general multi-SCC case via a recursive decomposition that uses interventional mean changes to recover SCC structure and topological order. Motivated by these results, the paper proposes a regularized least-squares estimator using both steady-state mean and covariance equations, and evaluates it on synthetic data and Perturb-seq datasets.

**Compliance With Llm Reviewing Policy:**

Affirmed.

**Final Justification:**

All concerns addressed, this is a good contribution

**Key Questions For Authors:**

1) Can the authors provide a more interpretable explanation of the spectral assumptions, or a stronger genericity result beyond numerical evidence?

2) How sensitive is the method to intervention mismatch, e.g. partial knockdown rather than exact removal of incoming regulation for a node?

3) Can the authors provide additional robustness analyses for misspecified non-diagonal diffusion and for finite-sample errors in recovering the SCC structure and topological order?

4) How sensitive are the results to estimator choices such as observational/interventional weighting, sparsity regularization, and covariance shrinkage?

**Limitations:**

yes

**Strengths And Weaknesses:**

Strengths

1) The main strength is the theoretical contribution. The shift from requiring many interventions to requiring only one intervention per SCC is meaningful and, to my knowledge, novel in this setting. The structural result that interventional mean changes reveal descendants of an SCC is elegant and gives the later recursive argument a clear causal interpretation. The multi-SCC recovery procedure is also more constructive than a purely existential identifiability statement, which improves the paper’s conceptual value.

2) A second strength is that the empirical section is aligned with the theory. On synthetic data, recovery of $\Lambda$ and $b$ improves as the number of interventions increases, while the covariance-only ablation performs substantially worse, supporting the claim that mean information is important. The real-data experiments also show that the proposed method can be competitive for unseen perturbation prediction on Perturb-seq benchmarks.

Weaknesses

1) My main concern is the narrowness of the modeling assumptions relative to the practical claims. The intervention model is specific: intervening on variable $i$ zeros all off-diagonal terms in row $i$ while preserving self-regulation, and diffusion is assumed diagonal. These assumptions are mathematically convenient, but the paper does not sufficiently discuss when they are realistic for biological perturbations or how robust the method is to approximate rather than exact interventions. Moreover, although the paper studies sensitivity to the number of interventions, graph density, and observational sample size, the robustness analysis remains incomplete relative to the strength of the assumptions. I would have liked to see additional sensitivity analyses for partial rather than exact interventions, misspecified non-diagonal diffusion, and key estimator hyperparameters such as observational/interventional weighting, sparsity regularization, and covariance shrinkage. These analyses would help clarify whether the method degrades gracefully under realistic departures from the assumed model.

2) The empirical evaluation is promising but still limited. On real data, the comparisons are mostly against ablations and simple baselines rather than a broader set of perturbation-prediction methods. Since one of the practical motivations is unseen perturbation prediction, stronger empirical comparisons would make the paper more convincing. In addition, the real-data gains are mixed rather than uniformly strong across DES and PDS.

overall:
I am positive on the theoretical core of the paper. The main identifiability result,namely, that one intervention per SCC can generically suffice to recover the OU parameters up to a global scaling under a connected SCC-DAG with a unique root and spectral nondegeneracy assumptions, is interesting, nontrivial, and a meaningful step beyond prior work that either relies only on observational stationary data or requires substantially stronger interventional coverage. I also found the recursive SCC-based decomposition conceptually appealing, since it gives a constructive route from interventional mean shifts to topological ordering and parameter recovery. That said, I think the paper is currently stronger as a theory paper than as a methods paper for practical perturbation modeling. The intervention model is highly specific, diagonal diffusion is assumed throughout, and the spectral conditions remain somewhat opaque without a stronger genericity result. In addition, while the empirical section is promising and aligns with the theory, it does not yet fully establish broad practical superiority: the real-data comparisons are relatively limited, the gains are mixed across DES and PDS, and I would have liked to see more robustness analysis under intervention misspecification and related forms of model mismatch. Overall, I lean positive because the identifiability contribution seems novel and worthwhile, but I would like to see a stronger discussion of assumption realism and more extensive empirical validation in a revision.

---

> ### Author Rebuttal · Authors · 2026-03-28
>
> > My main concern is the narrowness of the modeling assumptions...
>
> Regarding the intervention model, we emphasize that our formulation is consistent with standard notions of perfect interventions in causal SDEs. As stated in the paper (page 2, right column, lines 75–80), this is analogous to a knockout perturbation that isolates a variable from its regulators, and similar formulations have been considered in prior work (e.g., Hansen & Sokol, 2014; Boeken & Mooij, 2024).
>
> Regarding other types of interventions, some of our results can be extended. In particular, Theorem 3.3 and Remark 3.4 can be extended to soft interventions, where the i-th row is perturbed rather than exactly zeroed. In this case, we can still recover the SCC decomposition and the DAG over SCCs. We will add the corresponding proofs. We also performed a finite-sample experiment to evaluate SCC recovery from estimated means, using the reconstruction procedure of Theorem 3.3 and Remark 3.4 under both hard and soft interventions. SCC recovery was measured by the Adjusted Rand Index (ARI, where 1 indicates exact recovery), and the results show that recovery improves steadily as the number of samples increases.
>
> | Sample size | Hard int. (ARI) | Soft int. (ARI) |
> |-------------|--------------------------|--------------------------|
> | 500         | 0.660                    | 0.464                    |
> | 1000        | 0.746                    | 0.615                    |
> | 5000        | 0.849                    | 0.804                    |
> | 10000       | 0.881                    | 0.837                    |
>
> Regarding the diagonal diffusion assumption, it is mainly required for the full parameter identifiability result (Theorem 3.5). In fact, the learning objective in Eq. (10) does not necessarily require diagonal diffusion. We repeated the experiment in the same setup as Figure 1 with a non-diagonal diffusion matrix, and observed the same trend where relative error decreases consistently as the number of interventions increases.
>
> | #interventions | Rel. err. ($\Lambda$) | Rel. err. (D) | Rel. err. (b) |
> |----------------|--------------------------------|--------------------------|--------------------------|
> | 0              | 0.2064                         | 0.1811                   | 0.1407                   |
> | 2              | 0.1782                         | 0.1582                   | 0.1423                   |
> | 4              | 0.1570                         | 0.1449                   | 0.1394                   |
> | 6              | 0.1198                         | 0.1188                   | 0.1185                   |
> | 8              | 0.0859                         | 0.1011                   | 0.0994                   |
> | 10             | 0.0646                         | 0.0719                   | 0.0661                   |
>
> We will add the details of experiments and results in the revised version.
>
> On robustness and hyperparameters, we note that we already performed hyperparameter tuning for the obs./int. weighting and the sparsity regularization. We did not observe any significant degradation in performance over a relatively broad range of hyperparameter values in the loss function. We will add the details in the appendix.
>
> > The empirical evaluation is promising but still...
>
> Our main contribution is theoretical, namely providing identifiability results for linear SDEs from steady-state data (Theorems 3.1, 3.3, and 3.5). The empirical section is intended to illustrate the practical implications of these identifiability results (e.g., parameter recovery and intervention effects) rather than to benchmark a new learning algorithm. Nevertheless, as discussed in Section 6 (left column, lines 344–361), we focused on comparisons with methods that assume linear SDE models and operate on stationary data.
>
> > Q1: Can the authors provide a more interpretable...
>
> In Appendix C.2, we already provide an explanation of the role of the spectral assumptions in the proofs. These assumptions mainly serve to rule out degenerate algebraic cases and do not seem to admit a particularly direct interpretation. Similar technical nondegeneracy conditions also appear in other identifiability results, such as ANM in SCMs (see eq. (4) in Hoyer et al. 2009).
>
> > Q2: How sensitive is the method to intervention mismatch...
>
> As mentioned above, for recovering the SCC decomposition and the DAG over SCCs, our results can be extended to soft interventions. For full parameter identifiability, the hard intervention assumption is needed in our current proofs. The learning objective can also be adapted to soft intervention settings. If interventions perturb rows in a consistent way across variables, one can introduce shared parameters for this perturbation and jointly learn them together with other parameters.
>
> > Q3: Can the authors provide additional robustness analyses...
>
> We addressed both points above.
>
> > Q4: How sensitive are the results...
>
> We addressed it above

---

> > ### Author Rebuttal · Reviewer_GcTM · 2026-04-03
> >
> > Thank you, this is helpful clarification, i would be happy to raise my score to 5

---

### Decision · Program_Chairs · 2026-04-30

**Decision:**

Accept (spotlight)

**Comment:**

This work aims to identify the parameters of an OU process from stationary measurements. Typically such inference is impossible, however the authors show that in some cases, specifically under certain spectral nondegeneracy assumptions of the SCC condensation graph, the main parameters of the model can be recovered up to a scaling factor. This is a very interesting result both theoretically and practically as many biological measurement processes can only provide short, stationary measurements, requiring significant inference of the system dynamics. The authors follow up their theory with a real data example in genetics.

The theoretical results were especially appreciated by the reviewers as strong and nontrivial. Some reviewers thought the data example was a but lacking in completeness, however those were far outweighed by the overall strength and benefits of the work in general. More minor clarifications were primarily addressed in the rebuttal period. Therefore I recommend accepting this work.